# DSAS: A Universal Plug-and-Play Framework for Attention Optimization in Multi-Document Question Answering

**Jiakai Li[1], Rongzheng Wang[1], Yizhuo Ma[1],**
**Shuang Liang[1], Guangchun Luo[2], Ke Qin[1]***
[1]Institute of Intelligent Computing,
University of Electronic Science and Technology of China, Chengdu, China
[2]School of Information and Software Engineering,
University of Electronic Science and Technology of China, Chengdu, China
`ljk@std.uestc.edu.cn, qinke@uestc.edu.cn`

## Abstract

While large language models (LLMs) show considerable promise across various fields, they have notable limitations in handling multi-document question answering (Multi-doc QA) tasks. The first challenge is long-range dependency modeling, where LLMs struggle to focus on key information in long texts, which weakens important semantic connections. Second, most LLMs suffer from the "lost-in-the-middle" issue, where they have difficulty processing information in the middle of long inputs. Current solutions either truncate global dependencies or demand costly finetuning, ultimately lacking a universal and simple solution for these challenges. To resolve these limitations, we propose Dual-Stage Adaptive Sharpening (DSAS) containing two modules. (i) The Contextual Gate Weighting (CGW) module alleviates "lost-in-the-middle" by assessing paragraph relevance through layer-wise attention tracking and position-aware weighting. (ii) The Reciprocal Attention Suppression (RAS) module enhances focus on critical paragraphs by suppressing information exchange between key and irrelevant texts, thus mitigating the limitations in long-range dependency modeling. Extensive experiments on four benchmarks demonstrate DSAS's efficacy across mainstream LLMs (Llama, Qwen, Mistral, and Deepseek), with an average F1-score improvement of 4.2% in Multi-doc QA tasks on Llama-3.1-8B-Instruct and Qwen2.5-14B-Instruct. Ablation studies confirm the essential contributions of both the CGW and RAS modules. In addition, detailed discussions in the Appendix further validate the robustness and scalability of DSAS.

## 1 Introduction

Transformer-based [34] large language models (LLMs) have demonstrated remarkable performance, which have extensively promoted various complex natural language processing applications [9, 25, 30, 21, 37]. Building on the progress, recent advancements have shifted research focus toward enhancing LLMs' long-context processing capabilities [23], giving rise to LLMs that significantly expand context windows from 4K tokens to 128K or even 1M tokens, e.g., Llama-3.1-8B-Instruct [10] and Gemini-1.5 [31]. These LLMs have unlocked unprecedented potential for complex tasks requiring cross-document reasoning, such as legal case analysis [8] and multi-source academic synthesis [7].

---

*Corresponding author.
Code is available at `https://github.com/TreasureHunter/DSAS`

39th Conference on Neural Information Processing Systems (NeurIPS 2025).

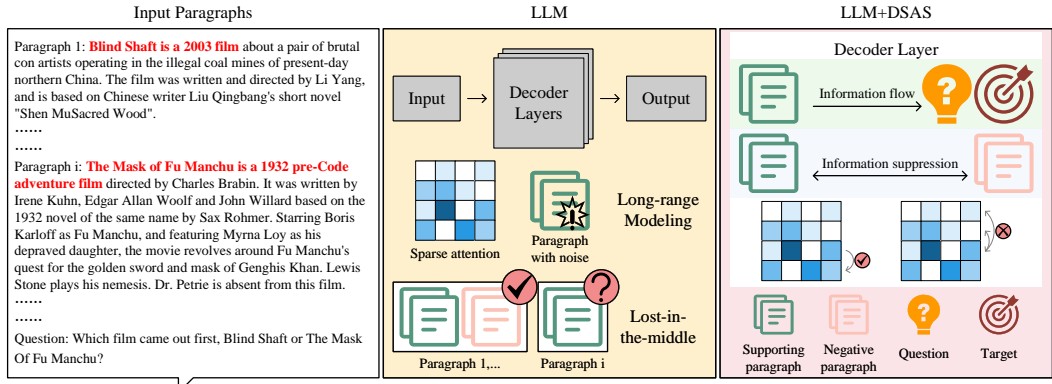

Figure 1: In Multi-doc QA tasks, directly processing long inputs comprising numerous paragraphs with LLMs presents two major challenges: long-range dependency modeling and "lost-in-the-middle", resulting in degraded answer quality. DSAS acts as a plug-in that enhances LLMs through a dual-stage process: (i) reinforcing information flow between supporting passages and both the question and target, and (ii) suppressing interactions between supporting and negative paragraphs.

However, simply concatenating multiple documents into these long contexts often results in degraded performance due to attention dilution [27], a phenomenon where critical inter-document dependencies are overshadowed by irrelevant tokens. While solutions like StreamingLLM [40] and Selective Self-attention [44] are introduced, they either truncate global dependencies or lack generalizability, leaving the core challenge of context-aware attention prioritization unresolved.

As shown in Figure 1, the aforementioned attention dilution phenomenon reveals two critical limitations in multi-document question answering (Multi-doc QA) scenarios: (i) Limited long-range dependency modeling: Despite claims of 128K-token support, RULER [15] reveals that LLMs often struggle with real-world tasks requiring combinatorial reasoning. While recent efforts (e.g., StreamingLLM [40], LM-Infinite [13]) explore attention mechanism optimization to address the challenge, they sacrifice global token interactions, compromising Multi-doc QA performance. (ii) Persistent "lost-in-the-middle" issue: Nelson et al. [24] demonstrate that LLMs perform poorly when key information appears in the middle of long inputs. Current solutions like LongAlign [2] using a hybrid strategy (combining long-instruction examples with short data) require additional training using curated datasets, making them less adaptable to mainstream LLMs. These challenges highlight the critical need for universal, plug-and-play modules that enhance Multi-doc QA capabilities without architectural constraints or task-specific fine-tuning.

Recent studies [35, 43, 17] demonstrate that attention-driven information flow analysis provides critical insights into model reasoning patterns. Inspired by this insight, we adapt the methodology to Multi-doc QA tasks. To address these issues, we introduce Dual-Stage Adaptive Sharpening (DSAS) as shown in Figure 1, a training-free attention optimization mechanism comprising two modules: Contextual Gate Weighting (CGW) and Reciprocal Attention Suppression (RAS). Specifically, our method works as follows: CGW tracks attention scores across selected model layers and calculates a contextual gate weight for each paragraph. Additionally, CGW introduces a position-aware weighting mechanism to enhance focus on information in the middle. Then RAS identifies critical paragraphs and attenuates information exchange between critical paragraphs and irrelevant content. DSAS requires no architectural changes or extra finetuning, serving as a universal plug-in for Transformer-based LLMs to strengthen Multi-doc QA capabilities.

Our contributions are as follows: (i) We systematically investigate the information flows on several LLMs in Multi-doc QA through paragraph disparity level and answer quality level analysis. (ii) Based on the findings, we propose DSAS, a training-free universal plug-in for Transformer-based LLMs, which enhances focus on critical information through CGW and RAS, while suppressing irrelevant content. (iii) Extensive experiments on four public benchmark datasets demonstrate the effectiveness of DSAS for various LLMs including Llama, Qwen, Mistral and Deepseek, achieving an average F1-score improvement of up to 4.2% on Llama-3.1-8B-Instruct [10] and Qwen2.5-14B-Instruct [28].

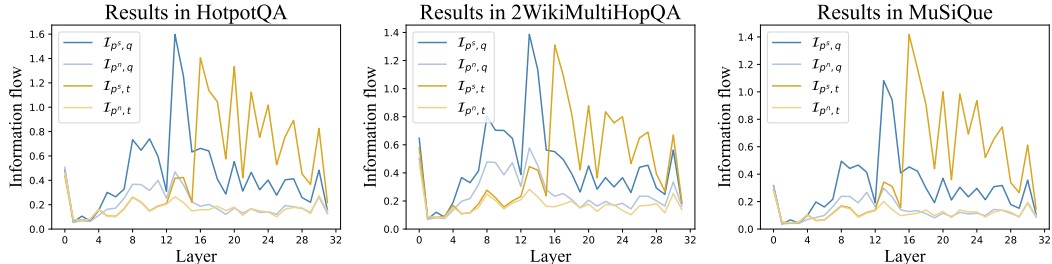

Figure 2: Layer-wise information flows of HotpotQA, 2WikiMultiHopQA and MuSiQue tested on Llama-3.1-8B-Instruct. $p^s$ and $p^n$ denote supporting paragraphs and negative paragraphs, respectively. The results of Qwen2.5-7B-Instruct are shown in Appendix A.2.

## 2 Information Flow Analysis on Multi-doc QA

Identifying key factors for effective Multi-doc QA reasoning is essential. To achieve this, we conduct a systematic analysis of LLMs' inference processes across three core components: input paragraphs ($p$), question ($q$), and target ($t$). A critical step lies in selecting suitable methods to study the semantic interactions among these components. Attention score analysis [29], a method widely used to examine information flow, is adapted here to investigate how LLMs integrate cross-document information.

### 2.1 Preliminaries

Let $A_{h,l}^S$, $A_{h,l}^W$ denote the attention score matrix and the attention weight matrix of the $h$-th head in the $l$-th layer, respectively. We obtain the layer-specific matrices $A_l^S$ and $A_l^W$ by summing across all attention heads. Here, $A_l^W(i,j)$ represents the information flow from the $j$-th to the $i$-th token.

To analyze interactions among three key components: (i) paragraphs ($p^1, \ldots, p^C$): Each paragraph $p^m$ spans input token indices $\{p_s^m, p_s^m + 1, \ldots, p_e^m\}$, where $p_s^m$ and $p_e^m$ denote the start/end positions and $C$ is the total paragraph count; (ii) question $q$; and (iii) target $t$: the answer generation position (i.e., the final token in the input), we propose two metrics. $\mathcal{I}_{p^m,q}$ and $\mathcal{I}_{p^m,t}$ are the Top-K significance of the information flow from the $m$-th paragraph $p^m$ to $q$ and $p^m$ to $t$, respectively:

$$\mathcal{I}_{p^m,q} = \frac{1}{Q} \sum \text{Top-K} \left( \left\{ \sum_{i \in q} A_l^W(i,j) | j \in p^m \right\} \right), \tag{1}$$

$$\mathcal{I}_{p^m,t} = \sum \text{Top-K} \left( \left\{ A_l^W(t,j) | j \in p^m \right\} \right), m \in \{1, \ldots, C\}, \tag{2}$$

where $Q$ denote the token length of the question. $\mathcal{I}_{p^m,q}$ and $\mathcal{I}_{p^m,t}$ serve as information flow indicators to analyze the inference process for LLMs. The larger their values are, the more the generated results pay attention to the corresponding paragraphs.

**Experimental Settings.** We choose Llama-3.1-8B-Instruct [10] and Qwen2.5-7B-Instruct [28] as our primary models for investigation, due to their moderate model size and strong instruction following ability. For datasets, since LongBench [3] lacks supporting facts (labels for supporting paragraphs), we use HotpotQA [41], 2WikiMultiHopQA [14] and MuSiQue [33]. We sample 1000 examples from the training set for evaluation. Templates for constructing inputs are provided in Appendix A.1.

### 2.2 Paragraph Disparity Level Analysis

Most LLMs adopt a multi-layer Transformer architecture, with each decoder block processing semantic information differently during inference. We intend to check layer-wise information flows of $\mathcal{I}_{p^m,q}$ and $\mathcal{I}_{p^m,t}$. Figure 2 visualizes layer-wise attention weight values. These values serve as a quantified metric to display how the different paragraphs contribute to the generated answer from the paragraph disparity perspective: Information flows from distinct paragraphs diverge only slightly in the shallow layers of the LLM, which suggests that the LLM first establishes basic semantic understanding to support deeper processing. As layers deepen, a clear divergence emerges between

$\mathcal{I}_{p^s,q}$ and $\mathcal{I}_{p^n,q}$, demonstrating the LLM progressively distinguishes task-relevant paragraphs through its layer-wise semantic processing, enabling rational attention allocation. Ultimately, in the deep layers of the LLM, LLM recognizes and utilizes key paragraphs for answer generation.

Our analysis uncovers a two-stage reasoning pattern: (i) Information initially converges on the question, with supporting paragraphs exhibiting stronger information flows than negative paragraphs, confirming the model's ability to prioritize semantically relevant content. (ii) Subsequently, the information from supporting paragraphs aggregates to the target, where the LLM strategically leverages critical paragraphs to formulate answers. These observations highlight the inherent interpretability of LLMs, proving their capabilities to integrate task-specific information during inference.

## 2.3 Answer Quality Level Analysis

We define a reasoning process as GOOD if its output exactly matches the reference answer, and BAD if it contains no word-level overlap with the reference answer (see Appendix A.4 for more details). To investigate the divergence between good and bad reasoning patterns, we aggregate information flow values across all model layers and conduct a comparative analysis of two distinct groups: $\mathcal{I}_{p^s,q}$ and $\mathcal{I}_{p^s,t}$ from supporting paragraphs,

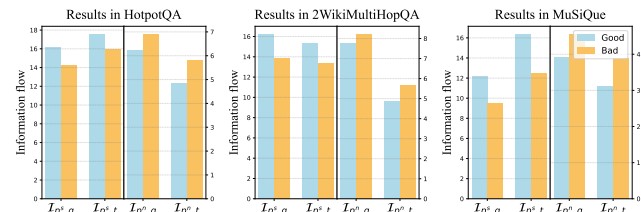

Figure 3: Comparison between mean values of the good and bad instances from the 1000 samples of Llama-3.1-8B. The results of Qwen2.5-7B are shown in Appendix A.3.

along with $\mathcal{I}_{p^n,q}$ and $\mathcal{I}_{p^n,t}$ from negative paragraphs. As shown in Figure 3, good reasoning exhibits higher values of $\mathcal{I}_{p^s,q}$ and $\mathcal{I}_{p^s,t}$, and lower values of $\mathcal{I}_{p^n,q}$ and $\mathcal{I}_{p^n,t}$. The MuSiQue dataset reveals the largest gaps between good and bad reasonings across all four metrics, which is probably attributable to its multi-hop reasoning complexity.

Notably, even in bad reasoning instances, $\mathcal{I}_{p^s,q}$ and $\mathcal{I}_{p^s,t}$ for supporting paragraphs consistently exceed those of $\mathcal{I}_{p^n,q}$ and $\mathcal{I}_{p^n,t}$ for negative paragraphs. This observation indicates that LLMs possess inherent discrimination capabilities between supporting and negative paragraphs regardless of reasoning quality, though the information flows of supporting paragraphs remain insufficient for optimal answer generation. Our framework addresses this core issue by introducing a quantitative information flow analysis that measures $\mathcal{I}_{p,q}$ and $\mathcal{I}_{p,t}$ for each paragraph. This analysis enables the selective and precise amplification of relevant semantic information, as detailed in Section 3.

## 3 Methodology

To explore the Multi-doc QA capabilities of LLMs, we analyze the attention score matrix in each layer and identify the key paragraphs through the information flows. Building on the above findings, we propose Dual-Stage Adaptive Sharpening (DSAS), which consists of two key modules. In Section 3.1, we introduce Contextual Gate Weighting (CGW), which identifies key paragraphs and strengthens the attention of the question and target position toward these texts. Section 3.2 describes Reciprocal Attention Suppression (RAS), which aims to suppress interactions between key paragraphs and irrelevant information. At a high level, we strategically integrate CGW and RAS modules into the computation of multi-head attention to implement DSAS:

$$
\begin{aligned}
A_{h,l}^S &= \frac{QW_{h,l}^Q(KW_{h,l}^K)^T}{\sqrt{d_k}}, \quad A_l^S = \text{Stack}(A_{1,l}^S, \ldots, A_{H,l}^S), \\
A_{h,l}^S &= \text{RAS}(\text{CGW}(A_{h,l}^S, A_l^S)), \\
A_{h,l}^W &= \text{Softmax}(A_{h,l}^S, \text{dim} = -1), \\
O_{h,l} &= A_{h,l}^W(VW_{h,l}^V), \quad O_l = \text{Concat}(O_{1,l}, \ldots, O_{H,l})W_l^O,
\end{aligned}
\tag{3}
$$

where $W_{h,l}^Q, W_{h,l}^K, W_{h,l}^V$ are projection matrices of the $h$-th head in the $l$-th layer, $W_l^O$ is the output projection matrix. We STACK tensors along the first dim and CONCAT them along the last dim. The framework of DSAS is shown in Figure 4. To avoid ambiguity, descriptions of key symbols appearing in the following text are provided in Table 5 in Appendix B.

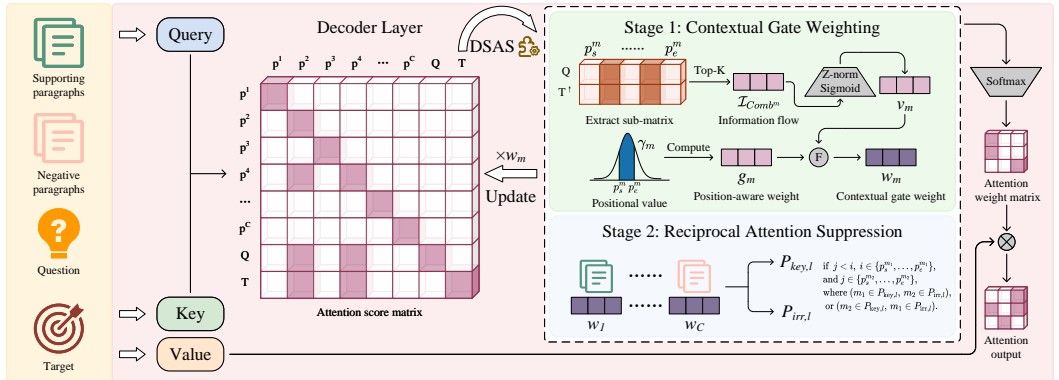

Figure 4: Illustration of Dual-Stage Adaptive Sharpening (DSAS), including Contextual Gate Weighting (CGW) and Reciprocal Attention Suppression (RAS).

## 3.1 Stage 1: Contextual Gate Weighting (CGW)

For each selected layer, we compute the combined information flows of each paragraph using the attention score matrix $A_l^S$. Specifically, we extract the attention sub-matrix corresponding to the positions of each paragraph, question and target as follows:

$$M \in \mathbb{R}^{2Q \times p^m} = \underbrace{[A_l^S(q, p^m)]}_{\text{Question Matrix}} \| \underbrace{[A_l^S(t, p^m) \uparrow^{(Q)}]}_{\text{Target Matrix}}, \tag{4}$$

where $Q, p^m$ denote the token length of the question and the $m$-th paragraph, respectively, and $\uparrow^{(Q)}$ represents the expand operation to match the question length. The combined information flow for the $m$-th paragraph is then calculated by averaging the column-wise Top-K values of $M$:

$$\mathcal{I}_{Comb^m} = \frac{1}{K} \sum \text{Top-K} \left[ \sum_{i=1}^{2Q} M_{i,j} \right]_{j=1}^{p^m}, m \in \{1, \ldots, C\}. \tag{5}$$

We then compute $v_m$ through Z-normalization and sigmoid scaling:

$$v_m = 0.5 \cdot \sigma \left( \frac{\mathcal{I}_{Comb^m} - \mu_I}{\sigma_I} \right) + 0.5, m \in \{1, \ldots, C\} \tag{6}$$

where $\sigma(\cdot)$ is the sigmoid function and $\mu_I, \sigma_I$ are the mean and the standard deviation of $\mathcal{I}_{Comb}$. The minimum value of $v_m$ is set to 0.5 to prevent overlooking the paragraphs excessively. In addition, to mitigate the "lost-in-the-middle" issue, we introduce position-aware weighting that assigns greater weights to central key paragraphs. Hsieh et al. [16] attribute this limitation to the inherent U-shaped attention bias in LLMs, which places more emphasis on content at the beginning and end of the sequence. We correct this bias through the probability density function (PDF) of a Gaussian distribution, which is defined as follows:

$$f(x) = \frac{1}{\sigma_p \sqrt{2\pi}} \cdot \exp \left( -\frac{(x - \mu_p)^2}{2\sigma_p^2} \right), \quad F(x) = \int_{-\infty}^{x} f(t) \, \mathrm{d}t \tag{7}$$

where $\mu_p, \sigma_p$ are the mean and the standard deviation of the input token indices $\{0, 1, \ldots, L - 1\}$ (i.e., $\mu_p$ equals $0.5 \cdot (L - 1)$, $\sigma_p$ equals $\sqrt{\frac{L^2-1}{12}}$), and $F(x)$ is the cumulative distribution function (CDF) of a Gaussian distribution. Then the positional value $\gamma_m$ for the $m$-th paragraph is computed based on the token indices $\{p_s^m, p_s^m + 1, \ldots, p_e^m\}$ within the segment:

$$z_1 = \frac{p_s^m - \mu_p}{\sigma_p}, \quad z_2 = \frac{p_e^m - \mu_p}{\sigma_p}, \tag{8}$$

$$\gamma_m = \frac{F(z_2) - F(z_1)}{z_2 - z_1}, \tag{9}$$

where $z_1, z_2$ denote the normalized value of $p_s^m$ and $p_e^m$, respectively. Next, we rank paragraphs by $v_m$, compute position-aware weights for the top 50% paragraphs, and assign a value of 1 to the rest, aiming to prevent the model from attracting excessive attention to non-critical middle paragraphs. The position-aware weight $g_m$ is calculated as follows:

$$g_m = \begin{cases} \left(\frac{0.5C+1}{\text{rank}_m}\right)^{\gamma_m}, & \text{if } \text{rank}_m \leq 0.5C, \\ 1, & \text{otherwise.} \end{cases} \quad (10)$$

The final contextual gate weight $w_m$ is derived as:

$$w_m^{'} = v_m \cdot g_m{}^{\alpha}, \quad (11)$$

$$w_m = (1 - \beta)\frac{w_m^{'} - \min(w_i^{'})}{\max(w_i^{'}) - \min(w_i^{'})} + \beta, i \in \{1, \ldots, C\} \quad (12)$$

where hyperparameters $\alpha$ and $\beta$ balance positional and content relevance and determine the minimum value of $w_m$, respectively. The attention score matrix $A_{h,l}^S$ is dynamically adjusted by applying contextual gate weights of each paragraph, thereby refining the model's focus based on their importance:

$$A_{h,l}^S(i, j) = w_m \cdot A_{h,l}^S(i, j), \quad \text{if } i \in \{q, t\}, j \in \{p_s^m, p_s^m + 1, \ldots, p_e^m\} \quad (13)$$

## 3.2 Stage 2: Reciprocal Attention Suppression (RAS)

The CGW module computes contextual gate weights for each paragraph to evaluate their relevance to answer generation, enabling clear distinction between critical and irrelevant paragraphs. The RAS module aims to suppress interactions between non-critical and key paragraphs. To achieve this, we classify paragraphs using a threshold-based method: paragraphs with contextual gate weights higher than the mean value of all $w_m$ are identified as key paragraphs, while those below the threshold are labeled as irrelevant paragraphs. The key and irrelevant paragraphs are denoted as $P_{key,l}$ and $P_{irr,l}$ in the $l$-th layer. Next, reciprocal attention suppression is applied between these categories. This process adjusts the attention score matrix $A_{h,l}^S$ by suppressing values between key and irrelevant paragraphs. The reciprocal suppression is bidirectional, affecting both interactions between $P_{key,l}$ and $P_{irr,l}$ to break cross-paragraph interference. Formally, the suppression is implemented as:

$$\begin{aligned} & A_{h,l}^S(i, j) = \min(w_{m_1}, w_{m_2}) \cdot A_{h,l}^S(i, j), \\ & \text{if } j < i, \ i \in \{p_s^{m_1}, p_s^{m_1}+1, \ldots, p_e^{m_1}\}, \text{and } j \in \{p_s^{m_2}, p_s^{m_2}+1, \ldots, p_e^{m_2}\}, \\ & \text{where } (m_1 \in P_{\text{key},l}, \ m_2 \in P_{\text{irr},l}) \text{ or } (m_2 \in P_{\text{key},l}, \ m_1 \in P_{\text{irr},l}). \end{aligned} \quad (14)$$

# 4 Experiments

This section assesses DSAS, a plug-and-play attention mechanism designed to enhance Multi-doc QA performance across diverse models and downstream tasks. Our experiments are structured as follows: Section 4.1 details the implementation of DSAS, covering datasets, metrics, models and hyperparameter settings. In Section 4.2, we show that DSAS achieves improvements on all Multi-doc QA benchmarks, including HotpotQA [41], 2WikiMultiHopQA [14], MuSiQue [33], and LongBench [3], without requiring any additional training. Section 4.3 presents ablation studies to examine the effectiveness of DSAS under different variants and hyperparameters. Section 4.4 further explores the robustness of DSAS.

## 4.1 Implementation

**Datasets & Metrics.** We evaluate our method on three classic Multi-doc QA benchmarks and a long-context dataset. Following previous works, we utilize the validation splits of HotpotQA [41], 2WikiMultiHopQA [14], and MuSiQue [33] to test our effectiveness, as they provide reference answers. To further assess generalization, we extend experiments to LongBench's [3] corresponding subsets on Multi-doc QA tasks. Referring to HotpotQA and LongBench, we consider the F1-score as the evaluation metric for these datasets. The metrics details are shown in Appendix A.4.

**Baselines & Models.** PINE [38] analyzes attention patterns to re-assign input paragraph positions. Given its similar operation and target to DSAS, we include PINE as a baseline for comparison, alongside a vanilla LLM baseline that performs inference without altering the original pipeline. We select

Table 1: Comparsion results on HotpotQA, 2WikiMultiHopQA, MuSiQue and LongBench benchmarks. 2WikiMQA denotes 2WikiMultiHopQA. The evaluation for all tasks is assessed through the F1-score (%).

| Models | Methods | HotpotQA | 2WikiMQA | MuSiQue | LongBench | | | Average |
| --- | --- | --- | --- | --- | --- | --- | --- | --- |
| | | | | | HotpotQA | 2WikiMQA | MuSiQue | |
| Llama-3.2-3B | Vanilla | 39.1 | 39.4 | 20.8 | 41.9 | 34.1 | 12.7 | 31.3 |
| | PINE | 39.0 | 39.6 | **21.3** | 42.4 | 35.3 | 12.5 | 31.6 |
| | DSAS | **40.6** (+1.5) | **39.7** (+0.3) | 21.2 (+0.4) | **42.7** (+0.8) | **35.9** (+1.8) | **13.2** (+0.5) | **32.2** (+0.9) |
| Mistral-7B | Vanilla | 32.0 | 19.0 | 24.3 | 39.3 | 16.6 | 22.2 | 25.6 |
| | PINE | 32.4 | 19.5 | 24.6 | 40.2 | **16.9** | 24.1 | 26.3 |
| | DSAS | **33.8** (+1.8) | **20.2** (+1.2) | **26.8** (+2.5) | **41.9** (+2.6) | 16.6 (+0) | **25.3** (+3.1) | **27.4** (+1.8) |
| Qwen2.5-7B | Vanilla | 42.3 | 46.0 | 30.5 | 55.1 | 50.3 | 28.1 | 42.1 |
| | PINE | 44.5 | 47.2 | 31.4 | 57.3 | 52.3 | 30.4 | 43.9 |
| | DSAS | **46.1** (+3.8) | **49.9** (+3.9) | **35.0** (+4.5) | **57.7** (+2.6) | **52.7** (+2.4) | **33.4** (+5.3) | **45.8** (+3.7) |
| DeepSeek-R1-8B | Vanilla | 37.2 | 22.9 | 26.8 | 45.1 | 20.3 | 24.8 | 29.5 |
| | PINE | 36.8 | 22.6 | 26.5 | 45.2 | 20.1 | 23.9 | 29.2 |
| | DSAS | **39.0** (+1.8) | **23.4** (+0.5) | **28.0** (+1.2) | **47.2** (+2.1) | **21.2** (+0.8) | **26.7** (+1.9) | **30.9** (+1.4) |
| Llama-3.1-8B | Vanilla | 43.6 | 47.3 | 34.6 | 53.3 | 42.6 | 25.4 | 41.1 |
| | PINE | 46.6 | 49.2 | 37.0 | 54.4 | 46.9 | 30.6 | 44.1 |
| | DSAS | **47.1** (+3.5) | **50.8** (+3.5) | **39.2** (+4.6) | **56.5** (+3.2) | **47.3** (+4.7) | **32.0** (+6.6) | **45.5** (+4.2) |
| Qwen2.5-14B | Vanilla | 48.2 | 55.3 | 38.0 | 57.6 | 53.1 | 32.8 | 47.5 |
| | PINE | 50.4 | 57.0 | 41.8 | 59.8 | 56.4 | 36.7 | 50.4 |
| | DSAS | **51.8** (+3.6) | **58.2** (+2.9) | **43.8** (+5.8) | **60.9** (+3.3) | **56.1** (+3.0) | **39.3** (+6.5) | **51.7** (+4.2) |
| Qwen2.5-32B | Vanilla | 48.8 | 60.7 | 42.3 | 58.6 | 48.2 | 35.0 | 48.9 |
| | PINE | 50.8 | 61.0 | 44.5 | 58.1 | 47.9 | 36.4 | 49.8 |
| | DSAS | **50.8** (+2.0) | **62.2** (+1.5) | **45.4** (+3.1) | **59.5** (+0.9) | **50.5** (+2.3) | **39.9** (+4.9) | **51.4** (+2.5) |

six popular LLMs, including Llama-3.2-3B-Instruct [10], Mistral-7B-Instruct-v0.2 [1], Qwen2.5-7B-Instruct [28], DeepSeek-R1-Distill-Llama-8B [11], Llama-3.1-8B-Instruct [10], Qwen2.5-14B-Instruct [28], and Qwen2.5-32B-Instruct [28], since they are popular Transformer-based decoder-only LLMs, which are convenient for exploiting and analyzing inside architectures. All models are deployed with the "bfloat16" data format due to the balance between efficiency and performance. We set the generation mode to greedy-decoding for all methods with deterministic sampling parameters: do_sample=False, temperature=0, top_p=1, max_new_tokens=32 (follow the setting in LongBench [3]). This setting minimizes the impact of irrelevant confounders during inference, thereby ensuring that identical models and inputs always produce the same answers to fixed questions. All experiments are implemented using PyTorch 2.2.1, and executed on the CPU of two 32-core Intel(R) Xeon(R) @ 2.80GHz and GPU of $8\times$ NVIDIA A800. The codebase is compatible with Python 3.12, and computations were accelerated using CUDA 12.1.

**Hyperparameters.** The hyperparameters in Equations 5, 11, and 12 are set to $K = 10$, $\alpha = 1$, and $\beta = 0.7$ for all benchmarks and LLMs. DSAS is applied to the final 50% layers of all LLMs.

## 4.2 Main Results

Table 1 presents the main experimental results, revealing three principal insights: (i) Simply replacing the original attention module with our DSAS enhances LLMs' performance across all Multi-doc QA benchmarks without requiring additional training, achieving average F1-score gains between 0.9% and 4.2%. Llama-3.1-8B and Qwen2.5-14B achieve the most significant improvements, particularly on LongBench tasks. These results demonstrate that DSAS acts as a catalyst to enhance Multi-doc QA performance across diverse model architectures and task configurations. In contrast, although PINE [38] achieves performance gains under certain configurations, its improvements are not consistently observed. (ii) Performance improvements hold consistently across model scales ranging from 3B to 32B parameters, with medium-sized LLMs generally exhibiting greater performance gains (e.g., maximum improvement of Llama-3.1-8B and Qwen2.5-14B, followed by Qwen2.5-7B). We attribute the pattern to two main reasons. (a) Medium-sized LLMs (e.g., Llama-3.1-8B, Qwen2.5-14B) demonstrate adequate semantic understanding yet remain vulnerable to input noise, especially in long-context scenarios. DSAS overcomes this by analyzing information flows to precisely identify critical paragraphs and amplify model focus. In contrast, smaller models (e.g., Llama-3.2-3B) lack sufficient comprehension capacity, while larger models (e.g., Qwen2.5-32B) approach the task performance ceiling, leaving little room for further gains. (b) Medium-sized LLMs exhibit greater vulnerability to

Table 2: Ablation study of DSAS on HotpotQA, 2WikiMultiHopQA, MuSiQue and LongBench benchmarks. 2WikiMQA denotes 2WikiMultiHopQA. p-a denotes position-aware weight $g_m$ in Equation (10). The evaluation for all tasks is assessed through the F1-score (%).

| Models | Variants | HotpotQA | 2WikiMQA | MuSiQue | LongBench | | | Average |
| --- | --- | --- | --- | --- | --- | --- | --- | --- |
| | | | | | HotpotQA | 2WikiMQA | MuSiQue | |
| Llama-3.2-3B | DSAS | **40.6** | **39.7** | 21.2 | 42.7 | **35.9** | **13.2** | **32.2** |
| | w/o CGW | 40.6 | 39.5 | 20.4 | **43.5** | 34.6 | 12.9 | 31.9 |
| | w/o RAS | 39.7 | 39.0 | **21.7** | 42.9 | 34.6 | 11.4 | 31.6 |
| | w/o p-a | 39.1 | 36.8 | 19.0 | 42.4 | 34.5 | 10.6 | 30.4 |
| Qwen2.5-7B | DSAS | **46.1** | 49.9 | **35.0** | **57.7** | **52.7** | **33.4** | **45.8** |
| | w/o CGW | 45.5 | 48.2 | 33.8 | 56.6 | 50.3 | 31.1 | 44.3 |
| | w/o RAS | 45.4 | 47.8 | 32.6 | 56.7 | 51.3 | 32.0 | 44.3 |
| | w/o p-a | 44.7 | **50.5** | 33.2 | 56.2 | 49.6 | 30.6 | 44.1 |
| Llama-3.1-8B | DSAS | **47.1** | **50.8** | **39.2** | **56.5** | 47.3 | **32.0** | **45.5** |
| | w/o CGW | 46.6 | 48.9 | 38.4 | 55.6 | 46.6 | 30.9 | 44.5 |
| | w/o RAS | 45.7 | 49.0 | 38.7 | 54.9 | 46.8 | 31.4 | 44.4 |
| | w/o p-a | 45.9 | 49.7 | 37.8 | 56.2 | 46.2 | 30.4 | 44.4 |
| Qwen2.5-14B | DSAS | **51.8** | 58.2 | **43.8** | 60.9 | **56.1** | **39.3** | **51.7** |
| | w/o CGW | 49.5 | 57.2 | 40.8 | 58.6 | 55.3 | 38.2 | 50.0 |
| | w/o RAS | 50.2 | 57.6 | 41.4 | 59.9 | 54.1 | 38.6 | 50.3 |
| | w/o p-a | 51.3 | **58.9** | 43.1 | **61.2** | 55.6 | 39.2 | 51.6 |

the "lost-in-the-middle" phenomenon, while larger models are generally more robust to this issue. Overall, DSAS effectively unlocks the latent capabilities of diverse LLMs on Multi-doc QA tasks, regardless of their architecture or scale. (iii) The improvement varies across different benchmarks. For medium-sized LLMs (parameters 7B, 8B, and 14B), greater improvements emerge on MuSiQue (including its LongBench extensions) compared to other benchmarks. This discrepancy suggests that DSAS particularly enhances LLMs' performance on complex tasks, where their inherent Multi-doc QA abilities can be better activated. The performance gap between HotpotQA and L-HotpotQA of these LLMs may be attributed to two factors. (a) Ansong et al. [26] identify annotation inconsistencies in HotpotQA that could result in unreliable assessments. (b) HotpotQA and L-HotpotQA only involve two-hop reasoning tasks with relatively lower complexity which are easy for these LLMs.

## 4.3 Ablation Studies

In this section, we conduct ablation studies to evaluate the effectiveness of each component in DSAS. Table 2 summarizes the performance of different variants on all tasks.

**w/o CGW.** Applying contextual gate weight $w_m$ only in RAS to regulate paragraph interactions degrades all metrics. This indicates that enhancing information propagation from key paragraphs to both the question and the target is essential for answer generation.

**w/o RAS.** Results reveal that removing $w_m$ during information aggregation lowers LLMs' performance, suggesting that the absence of RAS allows irrelevant content in $P_{irr,l}$ to introduce noise into the semantics of key paragraphs $P_{key,l}$.

**w/o p-a weight.** Setting $\alpha = 0$ in Equation (11) weakens resistance to the "lost-in-the-middle" issue, confirming that adaptively weighting middle paragraphs through $g_m$ enhances answer quality on most benchmarks. Notably, Qwen2.5-14B with this configuration achieves competitive results relative to DSAS, indicating larger LLMs inherently mitigate "lost-in-the-middle" more effectively, which aligns with the observations in LongPiBench [32]. Nevertheless, the position-aware weight generally improves performance on all benchmarks and LLMs, mitigating "lost-in-the-middle" issue.

**Hyperparameter Study.** The parameter $K$ in Equation (5) controls the number of tokens that the question and target attend to within each paragraph. Lower $K$ values risk insufficient contextual focus on the paragraphs, while higher values may degrade performance due to interference from irrelevant content. We evaluate $K$ with 5,10,20 and observe that $K = 10$ achieves optimal performance of Llama-3.1-8B, as shown in Figure 5 (a). Furthermore, we investigate the optimal number of insertion layers $n$ of DSAS. Section 2 shows minimal divergence in shallow layers. Most LLMs have layer counts in multiples of four (e.g., 28, 32). We therefore position DSAS at three types of depths: the final 25%, 50%, and 75% layers ($n = 25\%$, 50%, 75%, respectively) of the LLM. Figure 5 (b)

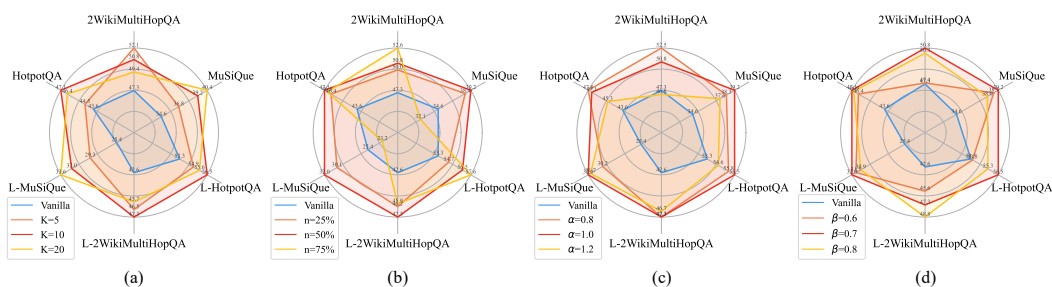

Figure 5: Llama-3.1-8B's hyperparameter study of $K$, $n$, $\alpha$, $\beta$ on HotpotQA, 2WikiMultiHopQA, MuSiQue, and LongBench benchmarks. The hyperparameter studies of Llama-3.2-3B, Qwen2.5-7B, and Qwen2.5-14B are shown in Appendix C.

Table 3: Results of original and shuffled input orderings when applying DSAS.

| Models | Variants | HotpotQA | 2WikiMQA | MuSiQue | LongBench | | | Average |
| --- | --- | --- | --- | --- | --- | --- | --- | --- |
| | | | | | HotpotQA | 2WikiMQA | MuSiQue | |
| Qwen2.5-7B | Original | **46.1** | **49.9** | 35.0 | **57.7** | 52.7 | **33.4** | 45.8 |
| | Shuffled | 46.1 | 49.7 | **35.4** | 57.6 | **53.0** | 33.3 | **45.9** |
| Llama-3.1-8B | Original | 47.1 | 50.8 | **39.2** | **56.5** | 47.3 | **32.0** | **45.5** |
| | Shuffled | **47.2** | **51.1** | 39.0 | 56.2 | **47.5** | 31.8 | 45.5 |
| Qwen2.5-14B | Original | **51.8** | 58.2 | **43.8** | 60.9 | **56.1** | **39.3** | **51.7** |
| | Shuffled | 51.7 | **58.3** | 43.8 | 60.9 | 56.0 | 39.3 | 51.7 |

demonstrates that best performance is achieved with DSAS at the final 50% layers position. This could be because $n = 25\%$ inadequately improves information flows, while $n = 75\%$ causes misjudgment of critical content due to the slight divergence among information flows of different paragraphs in LLMs' shallow layers. Finally, we discuss the most appropriate values for $\alpha$ in Equation (11) and $\beta$ in Equation (12), which are used to balance $v_m$ and $g_m$ and determine the minimum value of $w_m$, respectively. Figure 5 (c)(d) demonstrate $\alpha = 1$ and $\beta = 0.7$ are the best choices.

### 4.4 Further Analysis on Different Subsets

Since DSAS enhances attention on centrally located critical paragraphs, thereby mitigating the LLMs' inherent "lost-in-the-middle" issue. It is essential to evaluate its robustness to different input orders. To this end, we conduct an experiment in which we randomize the ordering of paragraphs in all test samples while deliberately increasing the probability of supporting paragraphs occurring at the edges. This shuffled configuration intentionally weakens the attention of DSAS to these important information. As shown in Table 3, results under both settings (using the fixed configuration $K = 10$, $n=50\%$, $\alpha = 1$, and $\beta = 0.7$) reveal only minor performance fluctuations across ordering conditions. This stability demonstrates the robustness of DSAS to input order, primarily benefitting from our weighting strategy. Specifically, when evidence appears at the edges (where $g_m$ in Equation (10) is reduced), DSAS attenuates the adjustments to the model's native attention distribution. This design intentionally leverages the model's inherent capacity to identify and amplify information flows from edge-positioned evidence, which aligns with the "U-shaped attention bias" phenomenon [16].

In addition, we further analyze how paragraph counts affect DSAS. Since the MuSiQue dataset contains 2-4 supporting paragraphs per sample, we group results by supporting paragraph count and report the comparative results in Table 4. We observe minimal performance gaps across the groups, indicating that DSAS better captures relevant information flows to improve accuracy. To examine the effect of total paragraph count, we select all samples from MuSiQue and construct inputs with 10, 20, 30, and 40 paragraphs. The 20-paragraph setting uses the original sample inputs; the 10-paragraph setting randomly removes ten of the negative paragraphs; the 30-paragraph and 40-paragraph settings add 10/20 paragraphs sampled from other examples. Table 4 shows that performance generally declines as the total paragraph count increases. However, DSAS degrades more slowly, since it suppresses information flow from negative paragraphs.

Table 4: Results on different subsets of MuSiQue. #Sup, #T denote supporting paragraph count and total paragraph count, respectively.

| Models | Variants | #Sup=2 | #Sup=3 | #Sup=4 | #T=10 | #T=20 | #T=30 | #T=40 |
|---|---|---|---|---|---|---|---|---|
| Qwen2.5-7B | Vanilla | 30.4 | 30.9 | 30.3 | 32.3 | 30.5 | 29.8 | 29.4 |
|  | DSAS | **35.0** | **35.2** | **34.7** | **36.7** | **35.0** | **34.8** | **34.3** |
| Llama-3.1-8B | Vanilla | 35.8 | 32.9 | 33.6 | 35.4 | 34.6 | 33.8 | 32.0 |
|  | DSAS | **40.2** | **35.7** | **36.5** | **39.5** | **39.2** | **38.7** | **38.3** |
| Qwen2.5-14B | Vanilla | 38.1 | 37.3 | 38.7 | 39.3 | 38.0 | 36.9 | 36.1 |
|  | DSAS | **44.0** | **43.3** | **44.3** | **44.7** | **43.8** | **43.5** | **42.9** |

## 5 Background and Related Works

**Long-context Reasoning in LLMs.** Recent advances in LLMs have spurred the development of long-context models (e.g., Llama-3.1-8B-Instruct [10] and Qwen2.5-7B-Instruct [28], claiming 128k-token capacities). However, mainstream capacity benchmarks like Needle-in-a-Haystack (NIAH) [19] mainly assess simple retrieval tasks, insufficiently reflecting complex real-world applications such as Multi-doc QA requiring cross-document information aggregation and multi-step reasoning. Current methods to enhance LLMs' long-context reasoning abilities follow two directions: (i) Positional encoding extensions (e.g., PI [6], CLEX [4], and CREAM [39]) effectively expand context windows by adjusting position embeddings, but demand extra training resources and struggle to integrate with existing LLMs with inherently large context windows. (ii) Attention mechanism optimizations reduce computational costs and support longer inputs with minimal additional training. However, most approaches prioritize efficiency over actual reasoning improvements. Both strategies leave significant gaps in enhancing LLMs' reasoning abilities within their designated context windows.

**Attention mechanism.** The Transformer [34] architecture LLMs currently face two challenges: quadratic computational and memory scaling with input length and the "lost-in-the-middle" [24] phenomenon. To overcome these constraints, recent studies are proposed to optimize the standard attention framework, including StreamingLLM [40], LM-Infinite [13], H2O [45], PINE [38], and SSC [42]. StreamingLLM and LM-Infinite optimize memory usage by retaining initial tokens and recent tokens for next-token prediction, balancing efficiency and performance. H2O preserves recent tokens while dynamically selecting critical tokens (termed heavy hitters) to balance local relevance and global importance. PINE employs bidirectional inter-segment attention and re-assigns paragraph positions. SSC scales hidden states to enable more balanced attention distribution across different segments. Collectively, current approaches either truncate long contexts to reduce computational and memory demands (e.g., StreamingLLM, LM-Infinite, H2O) or overlook the long-range modeling among paragraphs (e.g., PINE, SSC). Therefore, we propose DSAS, a training-free framework that adaptively optimizes attention matrices. Other approaches (e.g., prompt compression [18], retrieval-augmented generation [20], memory tree [5]) require external models or higher computation.

## 6 Conclusion

In this work, we address the critical challenges of limited long-range modeling and persistent "lost-in-the-middle" issue in Multi-doc QA for LLMs. Through a systematic analysis of information flow patterns from the paragraph disparity level and the answer quality level, our study reveals layer-wise aggregation patterns in distinct information flows and differences between good and bad reasoning instances. To resolve these issues, we propose Dual-Stage Adaptive Sharpening (DSAS), a training-free, plug-and-play mechanism that adaptively sharpens attention focus through two synergistic components: Contextual Gate Weighting (CGW) and Reciprocal Attention Suppression (RAS).

This work highlights the unexploited potential of attention optimization for LLMs. By adaptively sharpening attention score matrices, DSAS offers a practical solution for real-world applications requiring cross-document reasoning. Future directions include extending DSAS to diverse long-context scenarios, exploring its integration with retrieval-augmented generation frameworks (e.g., RAG [20], GraphRAG [9]), and investigating related LLM applications (i.e., backdoor attacks [12], data distillation [22], graph reasoning [36]). Our findings underscore the importance of optimal attention optimization in unlocking the full capabilities of modern LLMs for complex tasks.

## Acknowledgements

This work is supported by the National Natural Science Foundation of China (Grant No.62406057 and No.62176046), the Fundamental Research Funds for the Central Universities (Grant No.ZYGX2025XJ042), the Noncommunicable Chronic Diseases-National Science and Technology Major Project (Grant No.2023ZD0501806), and the Sichuan Science and Technology Program (Grant No.2024ZDZX0011). We gratefully acknowledge the Advanced Computing Frontier Research Center of Tianfu Jiangxi Laboratory for providing the computing resources that enabled this research, which has been successfully deployed in the laboratory's nurse agent.

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

# A Details of Information Flow Analysis on Multi-doc QA

## A.1 Experimental Templates

Consistent with LongBench [3], the templates for HotpotQA [41], 2WikiMultiHopQA [14], and MuSiQue [33] remain the same. Each data instance includes a question and its context, with task instructions positioned at both the start and end of the prompt to enhance model comprehension. This template is maintained consistently throughout our experiments in Section 4.

> Answer the question based on the given paragraphs. Only give me the answer and do not output any other words.
> The following are given paragraphs.
> {context}
> Answer the question based on the given paragraphs. Only give me the answer and do not output any other words.
> Question: {question}
> Answer:

## A.2 Paragraph Disparity Level Analysis on Qwen

Qwen2.5-7B consists of 28 stacked decoder layers. Analytical results on Figure 6 demonstrate similar conclusions to Llama-3.1-8B, with minimal information flow variations in shallow layers, followed by emerging divergences between $\mathcal{I}_{p^s,q}$ and $\mathcal{I}_{p^n,q}$, and finally gaps between $\mathcal{I}_{p^n,q}$ and $\mathcal{I}_{p^n,t}$. Notably, Qwen2.5-7B exhibits a slower progression of these divergences compared to Llama-3.1-8B, with the HotpotQA and the MuSiQue datasets particularly showing no significant information flow variations across the first 50% of the model's layers.

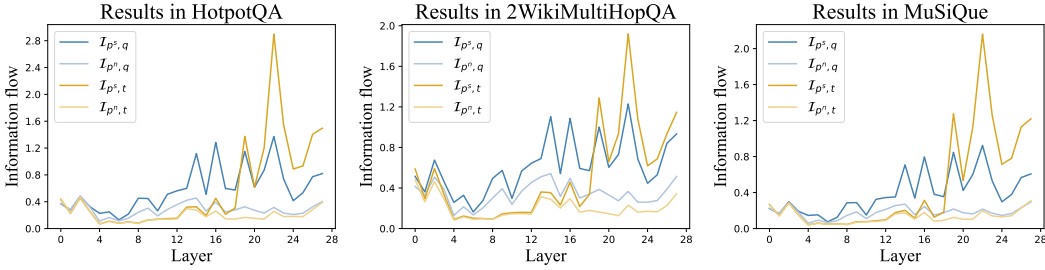

Figure 6: Layer-wise information flows of HotpotQA, 2WikiMultiHopQA and MuSiQue tested on Qwen2.5-7B. $p^s$ and $p^n$ denote supporting paragraphs and negative paragraphs, respectively.

## A.3 Answer Quality Level Analysis on Qwen

Similar to observations in Llama-3.1-8B, significant differences in information flow patterns emerge between good and bad reasoning instances, a consistent phenomenon observed in LLMs that forms the basis of DSAS framework.

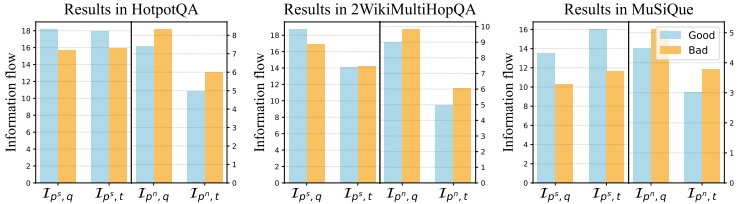

Figure 7: Comparison between mean values of the good and bad instances from the 1000 samples of Qwen2.5-7B.

### A.4 Answer Quality Assessment

The evaluation method employed in these Multi-doc QA benchmarks adopts the answer quality assessment approach from HotpotQA. Specifically, for each data instance, both the reference answer and response first undergo normalization, including lowercase conversion of all textual content, elimination of non-essential articles (i.e., a/an/the), and redundant whitespace. Then precision and F1-score are computed between the normalized reference answer and response through exact lexical matching. This assessment ensures comparability across different benchmarks. We classify samples achieving a perfect F1-score (equal to 1) as GOOD reasoning cases, and assign those with zero precision (equal to 0) to the BAD reasoning instances. The assessment method remains the same for experiments at Section 4.

## B  Symbol Explanation

Table 5: Symbols and Descriptions.

| Symbol | Description |
|---|---|
| $A_{h,l}^S$ | the attention score matrix of the $h$-th head in the $l$-th layer |
| $A_{h,l}^W$ | the attention weight matrix of the $h$-th head in the $l$-th layer |
| $\mathcal{I}_{comb^m}$ | the combined information flow for the $m$-th paragraph |
| $\mu_I, \sigma_I$ | the mean and the standard deviation of $\mathcal{I}_{comb}$ |
| $v_m$ | the normalized value of $\mathcal{I}_{comb^m}$ |
| $\mu_p, \sigma_p$ | the mean and the standard deviation of sequence $\{0, 1, \ldots, L-1\}$ |
| $\gamma_m$ | the positional value of the $m$-th paragraph |
| $g_m$ | the position-aware weight of the $m$-th paragraph |
| $w_m', w_m$ | the temporary and the final contextual gate weight of the $m$-th paragraph |
| $K$ | the number of tokens selected to compute $\mathcal{I}_{comb^m}$ |
| $n$ | the proportion of layers inserting DSAS |
| $\alpha, \beta$ | hyperparameters for computing $w_m$ |

## C  Full Hyperparameter Study

The results presented in Figures 8, 9, 10 indicate that DSAS maintains consistent improvement across diverse LLMs under the hyperparameter configuration in Section 4.1 (red regions in the radar charts). Furthermore, we observe that the robustness of DSAS to hyperparameter variations strengthens with LLMs' scale expansion (3B, 7B, 14B), with Qwen2.5-14B particularly showing notable and stable performance improvements across all benchmarks under different hyperparameter settings. We believe that this phenomenon correlates with LLMs' scales and their intrinsic capabilities. Additionally, some hyperparameter configurations of DSAS may marginally degrade performance below baseline for Llama-3.2-3B, which further validates that our method achieves maximum performance gains specifically for middle-sized LLMs (7B, 8B, 14B).

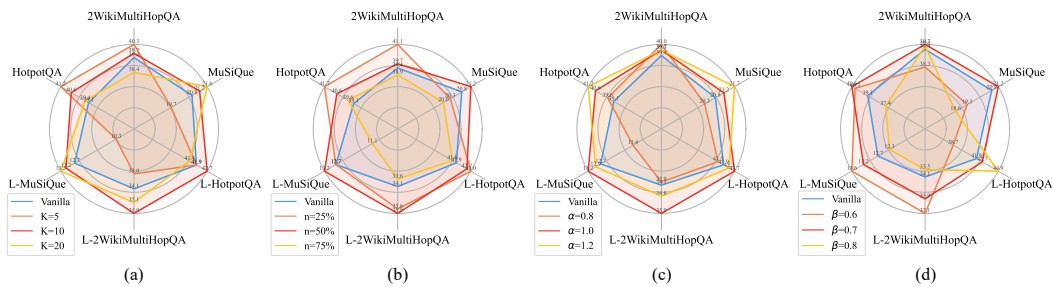

Figure 8: Hyperparameter study of $K$, $n$, $\alpha$, $\beta$ on HotpotQA, 2WikiMultiHopQA, MuSiQue, and LongBench benchmarks of Llama-3.2-3B.

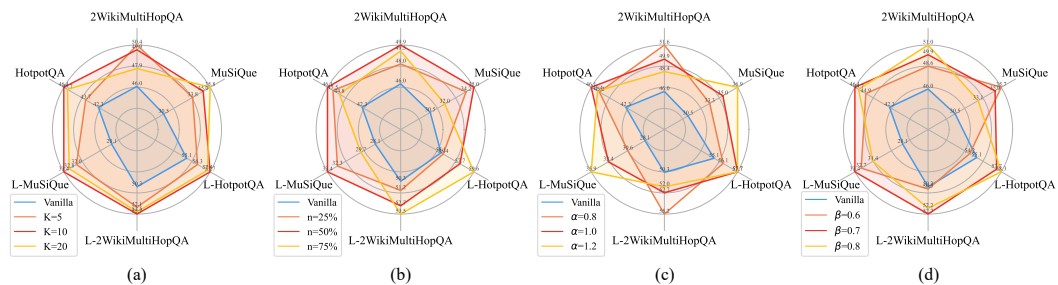

(a)       (b)       (c)       (d)

Figure 9: Hyperparameter study of $K$, $n$, $\alpha$, $\beta$ on HotpotQA, 2WikiMultiHopQA, MuSiQue, and LongBench benchmarks of Qwen2.5-7B.

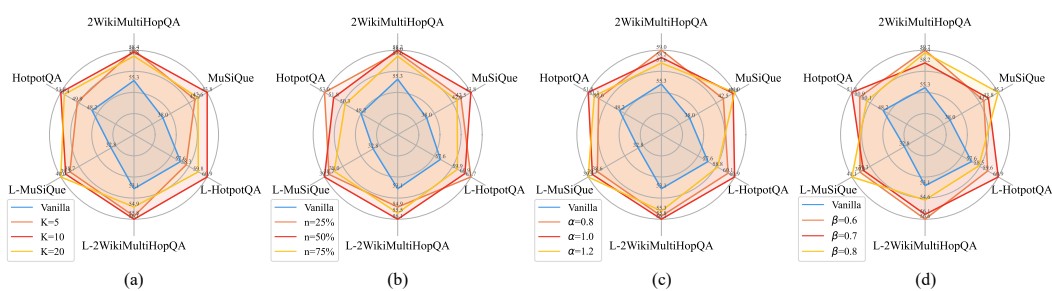

(a)       (b)       (c)       (d)

Figure 10: Hyperparameter study of $K$, $n$, $\alpha$, $\beta$ on HotpotQA, 2WikiMultiHopQA, MuSiQue, and LongBench benchmarks of Qwen2.5-14B.

# D   Error Analysis

We conduct an error analysis of DSAS with a focus on attention weight interactions among paragraphs (supporting and negative paragraphs), questions, and target. Following the experimental setup in Section 2, we randomly selected 1,000 samples for confusion matrix visualization between components. The analysis is conducted on the HotpotQA dataset due to its standardized structure of fixed 10-paragraph inputs (including 2 supporting paragraphs), which facilitates stable observation of interaction patterns.

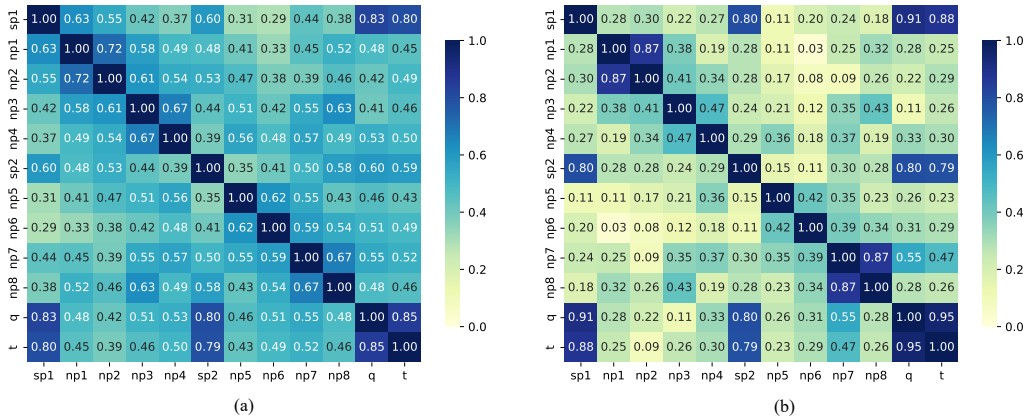

(a)       (b)

Figure 11: Confusion matrices on HotpotQA. (a) LLM, (b) LLM+DSAS. sp1, sp2 represent two supporting paragraphs. np1, np2, . . ., np8 represent eight negative paragraphs. q and t denote question and target, respectively. The results are conducted on Llama-3.1-8B-Instruct.

Technically, we first average attention weights across all layers to obtain a global attention matrix. For pairwise component analysis (e.g., between two paragraphs), we extract corresponding sub-matrix based on token index ranges. All component pairs are analyzed following this method. Similar to Equation 5, we average the column-wise Top-K values for each sub-matrix. Finally we normalize all confusion values to 0-1. This approach effectively captures salient attention patterns while minimizing noise interference. Figure 11 presents heatmap comparisons of reasoning processes using LLM and LLM+DSAS. The results demonstrate that LLM+DSAS not only strengthens the focus of questions and targets on relevant support paragraphs, but also enhances information interactions between these supporting paragraphs. Additionally, it effectively suppresses unnecessary interactions between supporting paragraphs and irrelevant negative paragraphs.

# E    Complexity Analysis

Table 6 presents the complexity of DSAS. Here, $L$ denotes the input token count, $Q$ denotes the question token count, and $C$ is the number of paragraphs. As illustrated in Equation 3, After computing the query, key, value matrices ($Q$, $K$, $V$) and attention scores $A_{h,l}^S$ through traditional attention mechanisms, we refine $A_{h,l}^S$ using the CGW and RAS modules. The updated scores are normalized via softmax to derive attention weights, which are then used to generate the final attention output. The core innovation of DSAS lies in computing contextual gate weight for dynamically weighting each paragraph. The time complexity scales linearly with $L$ and $Q$, contributing only a minor fraction of the total computation compared to the quadratic $\mathcal{O}(L^2)$ complexity of standard attention mechanisms. Furthermore, the space complexity grows linearly with $C$ alone. These characteristics ensure that DSAS maintains high efficiency in practice.

Table 6: Complexity analysis of DSAS.

| Time Complexity | $\mathcal{O}(LQ)$ |
|---|---|
| Space Complexity | $\mathcal{O}(C)$ |

# F    Scalability

Evaluating the scalability of DSAS beyond multi-document QA presents an interesting research direction. To this end, we apply DSAS to several long-context tasks from LongBench [3], including summarization (GovReport, QMSum, MultiNews) and code completion (LCC, RepoBench-P). The success of our information flow analysis largely relies on identifying anchors, which aggregate critical information from redundant contexts to guide answer generation. While in Multi-doc QA, the question and target naturally serve as anchors, extending DSAS to broader long-context tasks requires defining appropriate anchors for each scenario. After reviewing the prompt templates, we select the final instruction sentence or the query and the final sentence as anchors for aggregating information flows (e.g., "Now, write a one-page summary of the report.\n Summary:" for GovReport; "Next line of code:" for LCC).

Table 7: Comparison results on summarization and code completion tasks with hyperparameter configuration "$K$=10, $n$=50%, $\alpha$=1, $\beta$=0.7". The evaluation metrics for L-GovReport, L-QMSum, L-MultiNews and L-LCC, L-RBP are Rouge-L score and Edit Sim, respectively. RBP denotes RepoBench-P.

| Models | Variants | L-GovReport | L-QMSum | L-MultiNews | L-LCC | L-RBP | Average |
|---|---|---|---|---|---|---|---|
| Qwen2.5-7B | Vanilla | 33.6 | 22.4 | 23.7 | 53.7 | 48.2 | 36.3 |
| | DSAS | **37.0** | **25.1** | **26.2** | **56.3** | **51.5** | **39.2** |
| Llama-3.1-8B | Vanilla | 34.9 | 24.8 | 27.1 | 58.1 | 50.8 | 39.1 |
| | DSAS | **37.4** | **26.6** | **29.3** | **60.0** | **52.5** | **41.2** |
| Qwen2.5-14B | Vanilla | 38.4 | 28.2 | 31.8 | 63.9 | 55.6 | 43.6 |
| | DSAS | **41.0** | **30.4** | **34.6** | **66.7** | **58.9** | **46.3** |

Since DSAS operates on paragraph-level segments, which Multi-doc QA inherently provides, we employ fixed-length token segmentation for the other tasks. Specifically, we use 500-token segments for GovReport, QMSum, and RepoBench-P, and 200-token segments for MultiNews and LCC, ensuring a medium number (5-40) of paragraphs per example. The comparison results are presented in Table 7. Based on our experimental results, we make two key observations: (i) DSAS consistently enhances performance across all five tasks under three medium-sized model architectures; and (ii) these improvements generalize to diverse tasks (summarization and code completion), demonstrating that DSAS is effective beyond multi-doc QA.

The prompt templates used for these tasks align with those in the LongBench [3] benchmark. The sentences in red in the following template boxes serve as anchors.

The template for L-GovReport is shown below:

> You are given a report by a government agency. Write a one-page summary of the report.
> Report:
> {context}
> Now, write a one-page summary of the report.
> Summary:

The template for L-QMSum is shown below:

> You are given a meeting transcript and a query containing a question or instruction. Answer the query in one or more sentences.
> Transcript:
> {context}
> Now, answer the query based on the above meeting transcript in one or more sentences.
> Query: {input}
> Answer:

The template for L-MultiNews is shown below:

> You are given several news passages. Write a one-page summary of all news.
> News:
> {context}
> Now, write a one-page summary of all the news.
> Summary:

The template for L-LCC is shown below:

> Please complete the code given below.
> {context}
> Next line of code:

The template for L-RepoBench-P is shown below:

> Please complete the code given below.
> {context}{input}
> Next line of code:

# G  Limitations

While DSAS acts as a plug-and-play attention mechanism and consistently improves performance in Multi-doc QA tasks, our work has two primary limitations that warrant discussion. (1) The scalability of DSAS warrants further investigation. While our experiments on summarization and code completion tasks demonstrate consistent performance gains, the current fixed-token-count chunking strategy remains relatively simple. Future work could explore more refined and semantically-aware chunking methods for these tasks, which may further enhance information flow between key semantic segments and the generated answer. (2) Although DSAS enhances performance within standard LLM context windows, it inherits the fundamental computational and memory limitations of conventional Transformer [34] architectures, with quadratic complexity in both memory usage and

computational cost. Consequently, our approach does not address the scalability challenges associated with processing extremely long documents (e.g., those exceeding 100K tokens), where memory and computational demands become prohibitive. Future investigations could consider integrating sparse attention strategies or memory-efficient architectures to mitigate these constraints.

These limitations highlight important directions for future research while not diminishing the effectiveness of DSAS. The existing framework establishes a reliable base for advancing attention optimization in long-context processing.

