# OpenReview forum: "DSAS: A Universal Plug-and-Play Framework for Attention Optimization in Multi-Document Question Answering"
_NeurIPS.cc/2025/Conference — NeurIPS 2025 poster_

### Official Review · Reviewer_wMYQ · 2025-06-30

**Clarity:** 2
**Significance:** 3
**Originality:** 3
**Rating:** 3
**Confidence:** 4

**Summary:**

This paper proposes a plug-and-play method to improve LLM's performance on the Multi-doc QA task. They accumulate document-wise attention weights (they call it information flow) to inspect the attention patterns on document context in a multi-doc QA task. They get two observations: 1) support documents generally receive more attention, and 2) correct answers pay more attention to the support document compared to incorrect answers. Therefore, they propose a plug-and-play method to automatically detect support documents by accumulating attention weights and then reweighting supporting documents and negative documents by the accumulated attention, which upweighs the attention on supporting ones and downweighs the attention on negative ones. Otherwise, they propose a Gaussian-based score to reward middle-position documents to deal with the lost-in-the-middle problem. They also propose a Reciprocal attention suppression module, which penalizes the inter-document attention. They conducted experiments on several multi-doc QA datasets and showed that their methods can beat vanilla LLM without their method.

**Questions:**

Please refer to the questions in the "Strengths And Weaknesses" section.

**Ethical Concerns:**

["NO or VERY MINOR ethics concerns only"]

**Final Justification:**

I had a thorough discussion with the author of the paper. Given the typos and ambiguous symbols in important formulas and the omission of important baselines (which would require a complete rewrite of the entire experiments and related work sections), this paper requires significant revision for resubmission. Therefore, I give a score of 3 (Borderline reject).

**Limitations:**

yes

**Paper Formatting Concerns:**

No concerns

**Quality:**

2

**Strengths And Weaknesses:**

strengths:

i. This method is plug-and-play and easy to implement. I believe it will be very attractive and interesting if we can get performance improvement by simply manipulating the attention weights.

ii. This method is not only applicable to multi-document question answering, but also has the potential to be applied to RAG and general long-context generation tasks.

weakness:

i. I believe there are some mistakes in equations 7 and 8. E(*) is the cumulative function of the half-normal distribution with mu=0 and sigma=1/sqrt(2) with support R^+. But p_e - mu and p_s - mu do not guarantee to be positive. In Figure 4, it seems that you just want to use the cumulative distribution function of a general standard normal distribution. I do not understand why you write in this way. Additionally, in equation (8), it seems that you want to estimate \gemma by (F(x_2) - F(x_1))/2\sqrt(2)(x_2 - x_1) where x_2, x_1 is the normalized p_e and p_s, and F is the cumulative distribution function of N(0,1). This is naive because mu is set to the half-length and the derivative is high when x=0 (i.e., p_s and p_e near the middle). But I am confused about the discounting factor 2\sqrt(2), won't this make \gemma a quite small number and make g_m almost 1 everywhere?

ii. Ambiguous symbol usage, which is confusing. For example, A^{S}_{h,l} in equation 3, mu and sigma in equation 6 and equations 7&8. \omega_m in equation 11, A^{S}_{h,l} in 12.
quality:1

iii. Comparing only to one single baseline is not convincing enough. What are the implementation details of the only baseline method? Is it just vanilla LLM? Why do not compare with other baseline methods? I believe that there are several attention manipulation-based works in long-context generation or RAG, for example [1]. Are there any other baselines that can be implemented, such as masking documents lower than a threshold?

iv. In equation 1, you accumulate attention weights of question-length for each j without normalization, while in equation, there is only one attention weight for each j. However, in Figure 2, I_q and I_t show similar magnitudes. Why does this happen?

[1] Found in the Middle: Calibrating Positional Attention Bias Improves Long Context Utilization

---

> ### Author Rebuttal · Authors · 2025-07-31
>
> We sincerely appreciate the reviewer's thorough evaluation regarding equations, symbol definitions, and methodological comparisons.
>
> ***weakness i: Confusions about Equations***
>
> We welcome the opportunity to clarify Equations (7) and (8), which are defined in the manuscript as follows:
> $$
> \begin{align}
> F(x) &= \int\_{-\infty}^{x} f(t)\mathrm{d}t = 0.5 \left( 1 + E\left( \frac{x - \mu}{\hat{\sigma} \sqrt{2}} \right) \right), \quad
> E(z) = \frac{2}{\sqrt{\pi}} \int\_{0}^{z} e^{-t^{2}}\mathrm{d}t \\\\
> \gamma\_m&=\frac{\hat{\sigma}\left(E\left(\frac{p^m\_e-\mu}{\hat{\sigma}\sqrt{2}}\right)-E\left(\frac{p^m\_s-\mu}{\hat{\sigma}\sqrt{2}}\right) \right)}{2\left(p^m\_e-p^m\_s\right)}
> \end{align}
> $$
> (i) As stated in line 158 of this paper, $E(\*)$ is the **error function** and is used to define the CDF of the Gaussian distribution. While $E(\*)$ shares the same functional form as the CDF of the half-normal distribution (with $\mu$=0 and $\sigma=1/\sqrt 2$ with support $R^+$), it is defined over the entire real domain $R$.
>
> (ii) Equation (8) aims to compute a normalized difference quotient analogous to a derivative. Based on the derivation below, Equation (8) is actually equivalent to $\frac{F(z\_2)-F(z\_1)}{z\_2-z\_1}$, which lacks $2\sqrt 2$ in the denominator.
> $$
> \begin{align}
> Assume& \quad z\_1=\frac{p^m\_s-\mu}{\hat{\sigma}}, z\_2=\frac{p^m\_e-\mu}{\hat{\sigma}}. \\\\
> F(x)&=0.5\left( 1 + E\left( \frac{z}{\sqrt{2}} \right) \right). \\\\
> \gamma\_m&=\frac{\hat{\sigma}\left(E\left(\frac{p^m\_e-\mu}{\hat{\sigma}\sqrt{2}}\right)-E\left(\frac{p^m\_s-\mu}{\hat{\sigma}\sqrt{2}}\right) \right)}{2\left(p^m\_e-p^m\_s\right)} \\\\
> &=\frac{\hat{\sigma}\left(E\left( \frac{z\_2}{\sqrt{2}} \right)-E\left( \frac{z\_1}{\sqrt{2}} \right)\right)}{2((\hat{\sigma}z\_2+\mu)-(\hat{\sigma}z\_1+\mu))} \\\\
> &=\frac{F(z\_2)-F(z\_1)}{z\_2-z\_1}
> \end{align}
> $$
> (iii) When $p\_s$ and $p\_e$ reside near $\mu$ (i.e., the midpoint of the sequence), their normalized values $z\_1$ and $z\_2$ approach zero. Under these conditions, the resulting derivative $\gamma\_m$ (i.e., the positional value of the m-th paragraph) is high. The bigger $\gamma\_m$ value aligns with our core design objective: **Mitigate LLMs' "lost-in-the-middle" by enhancing attention weights assigned to centrally located information.**
>
> (iv) Based on the derivation of Equation (8) above, the denominator excludes the $2\sqrt 2$ factor. Consequently, this formulation does not make $\gamma\_{m}$ a quite small number.
>
> Should any uncertainties remain regarding this explanation, we welcome further discussion. To prevent confusion, we will enhance the textual descriptions of Equations (7) and (8) and add the complete derivation in the revised manuscript.
>
> ***weakness ii: Ambiguous Symbols***
>
> (i) $A^S\_{h,l}$ denotes the attention score matrix of the h-th head in the l-th layer. Equation (3) first computes the initial attention scores through dot products. We then apply both CGW and RAS to update its value. Our formulation follows the expressions in [1]. Equation (12) uses the calculated contextual gate weight of each paragraph to update $A^S\_{h,l}$.
>
> (ii) To resolve notational ambiguity, we clarify the distinct roles of $\mu$ and $\sigma$ across equations.
>
> $\mu$ in Equation (6) denotes the mean of combined information flow $\mathcal{I}\_{comb}$, defined in Line 153; $\mu$ in Equation (7) serves as a normalization value for scaling; $\mu$ in Equation (8) represents the mean of sequence $\{0,1,\ldots, L-1\}$, stated in Line 160.
>
> While $\sigma$ commonly denotes both sigmoid function and standard deviation, we use $\sigma$ for sigmoid function and $\hat{\sigma}$ for standard deviation in Equations (6)-(8) to disambiguate.
>
> However, the similar symbols may cause misunderstandings, so we revised Equations (6)-(8) to make them clearer:
> $$
> \begin{align}
> v\_m &= 0.5\cdot \text{Sigmoid}\left( \frac{\mathcal{I}\_{Comb^m} - \mu\_I}{\sigma\_I} \right)+0.5,m \in \{1,\ldots,C\}, \\\\
> F(x) &= \int\_{-\infty}^{x} f(t) \, \mathrm{d}t = 0.5 \left( 1 + E\left( \frac{x - \mu}{\sigma \sqrt{2}} \right) \right), \quad
> E(z) = \frac{2}{\sqrt{\pi}} \int\_{0}^{z} e^{-t^{2}} \, \mathrm{d}t, \\\\
> \gamma\_m&=\frac{\sigma\_p\left(E\left(\frac{p^m\_e-\mu\_p}{\sigma\_p\sqrt{2}}\right)-E\left(\frac{p^m\_s-\mu\_p}{\sigma\_p\sqrt{2}}\right) \right)}{2\left(p^m\_e-p^m\_s\right)},
> \end{align}
> $$
> (iii) To explicitly combines $v\_m$ and $g\_m$ while constraining $w\_m$ within $[\beta,1]$, we introduce an intermediate variable $w\_m^{'}$ for Equations (10) and (11):
>
> $$
> \begin{align}
>     w\_m^{'}&=v\_m \cdot {g\_m}^\alpha, \\\\
>     w\_m&=(1-\beta)\frac{w\_m^{'}-\text{min}(w\_i^{'})}{\text{max}(w\_i^{'})-\text{min}(w\_i^{'})}+\beta,i \in \{1,\ldots,C\}
> \end{align}
> $$
>
> (iv) Lastly, To mitigate potential notational ambiguities, we will add Table 1 of key symbols in Appendix.
>
> Table 1 Symbols and descriptions.
>
> | Symbol | Description |
> | --- | --- |
> | $A^S\_{h,l}$ | the attention score matrix of the h-th head in the l-th layer |
> | $A^W\_{h,l}$ | the attention weight matrix of the h-th head in the l-th layer |
> | $\mathcal{I}\_{{comb}^m}$ | the combined information flow for the m-th paragraph |
> | $\mu\_I, \sigma\_I$  | the mean and the standard deviation of $\mathcal{I}\_{comb}$ |
> | $v\_m$  | the normalized value of $\mathcal{I}\_{{comb}^m}$ |
> | $\mu\_p, \sigma\_p$ | the mean and the standard deviation of sequence $\{0,1,\ldots,L-1\}$ |
> | $\gamma\_m$ | the positional value of the m-th paragraph|
> | $g\_m$ | the position-aware weight of the m-th paragraph |
> | $w\_m^{'}, w\_m$ | the temporary and the final contextual gate weight of the m-th paragraph |
> | $K$  | the number of tokens selected to compute $\mathcal{I}\_{{comb}^m}$ |
> | $n$  | the proportion of layers inserting DSAS |
> | $\alpha, \beta$ | hyperparameters for computing $w\_m$ |
>
> [1] Xiao, Da, et al. "Improving transformers with dynamically composable multi-head attention." arXiv preprint arXiv:2405.08553 (2024).
>
> ***weakness iii: Comparisons of Different Methods***
>
> (i) As described in line 205 of the manuscript, our baseline implementation deploys all LLMs in the ''bfloat16'' data format and employs greedy-decoding to ensure full reproducibility of generated outputs.
>
> (ii) The Calibrated attention proposed in [2] shares a similar motivation with our approach in seeking to enhance LLMs' focus on central information within long contexts. **However, DSAS extends this objective by actively promoting information flows from supporting paragraphs while suppressing information flows from irrelevant information.** Specifically, Calibrated Attention quantifies the model's positional bias by fixing a query-paragraph pair and measuring how the self-attention distribution changes as the paragraph is relocated within the input. During inference, it subtracts the measured positional bias from the original attention scores to obtain calibrated attention weights. One limitation is its requirement to precompute the positional bias for each paragraph, which may entail significant computational overhead. Since the implementation of [2] is not available, we reproduced a version guided by its core methodology. The results are summarized in Table 2.
>
> (iii) To our knowledge, existing attention efforts mainly concentrate on sparse attention mechanisms [3], context compression [4], and KV cache optimization [5], while research addressing performance improvements in long-context generation remains underexplored.
>
> Table 2 Comparison results of different methods.
>
> | Models       | Method    | HotpotQA | 2WikiMultiHopQA | MuSiQue | L-HotpotQA | L-2WikiMultiHopQA | L-MuSiQue |
> | :-: | :-: | :-: | :-: | :-: | :-: | :-: | :-: |
> | Qwen-2.5-7B  | Calibrated | 44.5 | 48.6 | 33.6 | 57.1 | 51.6 | 30.3 |
> | | Ours       | **46.1** | **49.9** | **35.0** | **57.7** | **52.7** | **33.4**  |
> | Llama-3.1-8B | Calibrated | 46.4 | 50.0 | 37.3 | 55.2 | 46.5 | 28.6 |
> | | Ours       | **47.1** | **50.8** | **39.2** | **56.5** | **47.3** | **32.0**  |
> | Qwen-2.5-14B | Calibrated | 50.4 | 57.1 | 40.4 | 58.1 | 54.9 | 36.4 |
> | | Ours       | **51.8** | **58.2** | **43.8** | **60.9** | **56.1** | **39.3**  |
>
> [2] Hsieh, Cheng-Yu, et al. "Found in the middle: Calibrating positional attention bias improves long context utilization." arXiv preprint arXiv:2406.16008 (2024).
>
> [3] Jiang, Huiqiang, et al. "Minference 1.0: Accelerating pre-filling for long-context llms via dynamic sparse attention." arXiv preprint arXiv:2407.02490 (2024).
>
> [4] Zhang, Peitian, et al. "Long context compression with activation beacon." arXiv preprint arXiv:2401.03462 (2024).
>
> [5] Qin, Ziran, et al. "Cake: Cascading and adaptive kv cache eviction with layer preferences." arXiv preprint arXiv:2503.12491 (2025).
>
> ***weakness iv: Definitions of Equations***
>
> We thank the reviewer for raising this point and apologize for the oversight. Equation (1) computes attention weights summed across all tokens in the question (length $Q$). In contrast, Equation (2) utilizes the attention weight for a single target token $t$. To make the equations directly comparable, our code implementation averages the accumulated attention weights over the sequence length $Q$ in Equation (1). We will correct Equation (1) accordingly in the final version. Figures 3 and 7 remain correct with this revision. The updated Equation (1) is given below:
> $$
> \mathcal{I}\_{p^m,q} = \frac{1}{Q} \sum \text{Top-K} \left( \left\\{  \sum\_{i \in q} A\_l^W(i,j) | j \in p^m      \right\\} \right).
> $$

---

> > ### Comment · Reviewer_wMYQ · 2025-08-07
> > **discussion to 20041**
> >
> > Thank you for your rebuttal and newly added experiments.
> >
> >
> > I accept your explanation of equations (7) and (8) (The misunderstanding of equation 8 comes from a different definition of $z_2$ and $z_1$). However, there are still two concerns remaining:
> >
> >
> > - I do not understand the benefit of introducing the error function $E(*)$. Why not directly define F(x) as a general cdf of standard normal distribution $\int_{-\infty}^{x}\frac{1}{\sqrt{2\pi}\sigma}e^{-\frac{(x-\mu)^2}{2\sigma^2}}$, then equation 8 will be written directly as $\frac{F(z_2)-F(z_1)}{z_2-z_1}$. The introduction of the error function only brings confusion.
> >
> >
> > - as you wrote, $\gamma_m = \frac{F(z_2)-F(z_1)}{z_2-z_1}$ and $z_2 = \frac{p^m_e - \mu}{\sigma}$ and $z_1 = \frac{p^m_s - \mu}{\sigma}$. We know $p^m_s, p^m_e \in [0, L]$, $\mu = 0.5 (L-1)$, $\sigma = \frac{L^2-1}{12}$. Then we can easily derive the range of $z_2$ and $z_1$, i.e. $z_1, z_2 \in [-\frac{6}{L+1}, \frac{6}{L-1}]$. It's obvious that if $L$ is large (which is common because it is the input length of all passages), the distance between $z_2$ and $z_1$ is quite small for any pair of $(p^m_s, p^m_e)$, which leads to a similar value of $\gamma_m$. I made a naive trial with L=500, and the value of $(p^m_s, p^m_e) $and $\gamma_m$ is as follows:
> >   - $p^m_s = 400, p^m_e = 500\rightarrow\gamma_m=0.3989234$
> >   - $p^m_s = 100, p^m_e = 200\rightarrow\gamma_m=0.3989373$
> >   - $p^m_s = 200, p^m_e = 300\rightarrow\gamma_m=0.3989418$
> >   - $p^m_s = 240, p^m_e = 260\rightarrow\gamma_m=0.3989422$
> >
> >   Therefore, although "this formulation does not make $\gamma_m$ a quite small number.", it seems to make it indiscriminate.
> >
> > As for the weakness 3 "Comparisons of Different Methods" (1),  line 205 of the manuscript, "All the models are deployed with the “bfloat16” data format due to the balance between efficiency and performance." is ***NOT*** a description of a method.
> >
> > As for the weakness 3 "Comparisons of Different Methods" (2), my concern is not about a certain specific baseline (Found in the Middle: Calibrating Positional Attention Bias Improves Long Context Utilization).  My concern is that as a non-professional in this area, I can easily Google several comparable baselines within several minutes, such as [1][2]. But you neither compare with any of them, nor give an acceptable reason. I believe there is more research in this area. Therefore, I do ***NOT*** think that it is ok to claim that "research addressing performance improvements with attention efforts in long-context generation remains underexplored.". And more baseline methods are needed to prove the effectiveness of your method.
> >
> > [1] Wang et al. Eliminating Position Bias of Language Models: A Mechanistic Approach.
> > [2] Fang et al. AttentionRAG: Attention-Guided Context Pruning in Retrieval-Augmented Generation

---

> > > ### Author Response · Authors · 2025-08-08
> > >
> > > We thank the reviewer for the valuable feedback.
> > >
> > > ***Concern1: The Introduction of $E(\*)$***
> > >
> > > Our code implementation employs the error function $E(\*)$ for Gaussian distribution CDF computations due to its algorithmic superiority and efficiency, aligning with our paper's definition. We appreciate the reviewer for identifying a clearer and more concise formulation of Equation (8). We will update the formulations in the revised manuscript. Crucially, the different expressions are mathematically equivalent and do not affect the novelty of our paper.
> > >
> > > ***Concern2: The Computation of $\gamma\_m$***
> > >
> > > We correct a mathematical mistake in line 160 in the initial manuscript. For the input token indices $\\{0,1,\ldots,L-1\\}$, $\mu$ equals $0.5(L-1)$ and $\sigma$ equals $\sqrt \frac{L^2-1}{12}$, where **the square root operator was previously omitted**. Under this setting, we can roughly derive that $z\_1, z\_2 \in [-\sqrt 3,\sqrt 3]$ for large $L$ (e.g., 16k). When $L$ eqauls 500 (with indices from 0 to 499), where $\mu$ and $\sigma$ are 249.5 and 144.34, the resulting $\gamma\_m$ are as follows:
> > >
> > > * $p\_s^m=400,p_e^m=499 \to z\_1=1.0427, z\_2=1.7286 \to \gamma\_m=0.1554$
> > > * $p\_s^m=100,p_e^m=200 \to z\_1=-1.0357, z\_2=-0.3429 \to \gamma\_m=0.3113$
> > > * $p\_s^m=200,p_e^m=300 \to z\_1=-0.3429, z\_2=0.3499 \to \gamma\_m=0.3911$
> > > * $p\_s^m=240,p_e^m=260 \to z\_1=-0.0658, z\_2=0.0727 \to \gamma\_m=0.3988$
> > >
> > > We overlooked the notation error during manuscript checks. However, our code correctly follows the intended formulations, which assigns distinct $\gamma\_m$ values to paragraphs based on their positions, and all empirical results remain valid.
> > >
> > > ***Discussion1: The Description of Baseline LLM***
> > >
> > > (i) **The "Baseline LLM" is valinna LLM.**
> > >
> > > (ii) To ensure full reproducibility of outputs, the "Baseline LLM" configuration employs greedy-decoding with deterministic sampling parameters: do_sample=False, temparature=0, top_p=1, max_new_tokens=32 (follow the setting in LongBench [1]).
> > >
> > > (iii) All experiments were conducted under a zero-shot setting. As shown in Appendix A.1, the inputs follow a same template, including instructions, a question and the context. The inputs of "Baseline LLM" configuration are consistent with "LLM+DSAS". Due to the maximum input length is less than 32k, no truncation was employed, since all LLMs support $\ge$32k context length.
> > >
> > > (iv) The original outputs are the final answers. We calculated the F1-score for evaluation, which is detailed in Appendix A.4.
> > >
> > > [1] Bai, Yushi, et al. "Longbench: A bilingual, multitask benchmark for long context understanding." arXiv preprint arXiv:2308.14508 (2023).
> > >
> > > ***Discussion2: Comparisons with Peer Methods***
> > >
> > > **The comparative results will be reported after all experiments finished.**

---

> > > > ### Author Response · Authors · 2025-08-09
> > > >
> > > > ***Discussion2: Comparisons with Peer Methods***
> > > >
> > > > We appreciate the reviewer for highlighting relevant studies, such as \[1\]\[2\]. Through supplementary experiments, we have further validated the effectiveness of our method. Additionally, we have introduced new baselines \[3][4] based on similar thoughts.
> > > >
> > > > Specifically, Wang et al. [1] enhance focus on critical paragraphs by using bidirectional inter-segment attention and re-assigning positions of the paragraphs, while Fang et al. [2] analyze attention scores to concentrate on key paragraphs for effective prompt compression. Yu et al. [3] scale hidden states to enable more balanced attention distribution across different segments. PEAR [4] identifies attention heads that negatively affect RAG performance and learns parameters to reweight these head outputs.
> > > >
> > > > We concede that the paper overly focused on highlighting the generality and effectiveness of our method and weakened the comparisons with peer methods. **In fact, as shown in the following experiments, our method significantly outperforms several baselines, which proves our effectiveness.**
> > > >
> > > > The implementation of \[1]\[3]\[4] is accessible through the publicly available repositories released by their authors, and we report the reproduced results. The results of [2] are sourced from its paper. In addition, we further add the performance of our method under one-shot setting.
> > > >
> > > > Experimental setting: To ensure a fair comparison, we follow the same rules illustrated in ***Discussion1*** (ii)(iii)(iv) for all the methods, with consistent sampling parameters, prompt and evaluation metrics. We use the official script to train PEAR [4] only on Qwen2.5-7B due to time limit. All experiments are conducted on NVIDIA A800 GPUs. For the experiment of ''DSAS-1shot'', we add one example following the same template and one sentence "Do not use the information above to answer the following question.\n\n" before the zero-shot prompt. **The comparisons include [1], [2], [3], [4], ''DSAS'', ''DSAS-1shot''.**
> > > >
> > > > Table 1 Comparison results of different methods (F1-score).
> > > >
> > > > |    Models    |   Method   | L-HotpotQA | L-2WikiMultiHopQA | L-MuSiQue |
> > > > | :- | :- | :-: | :-: | :-: |
> > > > |  Qwen2.5-7B  |  PINE [1]  |    57.3    |       52.3        |   30.4    |
> > > > |  |  SPHS [3]  |    54.2    |       51.1        |   26.8    |
> > > > |  |  PEAR [4]  |    56.9    |     **53.4**      |   32.5    |
> > > > |  |    DSAS    |  **57.7**  |       52.7        | **33.4**  |
> > > > |  | DSAS-1shot |  **58.4**  |       52.9        | **34.1**  |
> > > > | Llama-3.1-8B |  PINE [1]  |    54.4    |       46.9        |   30.6    |
> > > > |  |  SPHS [3]  |    53.0    |       43.2        |   27.1    |
> > > > |  |    DSAS    |  **56.5**  |     **47.3**      | **32.0**  |
> > > > |  | DSAS-1shot |  **57.1**  |     **51.0**      | **32.7**  |
> > > >
> > > > Table 2 Comparison results of different methods (EM score). EM (Exact Match) score measures the percentage of predicted answers that exactly match the ground truth answers.
> > > >
> > > > | Models | Method | L-HotpotQA | L-2WikiMultiHopQA |
> > > > | :- | :- | :-: | :-: |
> > > > | Qwen-2.5-7B  | AttentionRag [2] |    53.0    |       41.0        |
> > > > |  | DSAS |  **54.5**  |     **48.0**      |
> > > > | Llama-3.1-8B | AttentionRag [2] |    48.0    |       42.0        |
> > > > |  |       DSAS       |  **54.0**  |     **44.5**      |
> > > >
> > > > Result analysis: PINE [1] locates critical paragraphs toward later input positions, while SPHS [3] selectively adjusts hidden states through specific channel identification. Both methods mitigate ''lost-in-the-middle'' and exhibit robustness to positional variance, but neither accounts for interactions among key paragraphs, which are crucial in complex tasks like Multi-doc QA. AttentionRag [2] improves inference efficiency by dropping irrelevant context, achieving poorer results. PEAR [4] requires additional training and shows limited generalizability. "DSAS-1shot" marginally outperforms "DSAS", which we attribute it to the more standardized outputs with fewer irrelevant texts.
> > > >
> > > > We appreciate your comments for our work again. **The introductions and relevant experiments of peer methods will be included in our revised paper.** As the discussion phase is approaching its end, we still kindly request you to let us know if the above clarifications have addressed your concerns. We would be happy to address any additional points you may have during the remaining time of the discussion phase.
> > > >
> > > > [1] Wang, Ziqi, et al. "Eliminating position bias of language models: A mechanistic approach." arXiv preprint arXiv:2407.01100 (2024).
> > > >
> > > > [2] Fang, Yixiong, et al. "Attentionrag: Attention-guided context pruning in retrieval-augmented generation." arXiv preprint arXiv:2503.10720 (2025).
> > > >
> > > > [3] Yu, Yijiong, et al. "Mitigate Position Bias in LLMs via Scaling a Single Hidden States Channel." arXiv preprint arXiv:2406.02536 (2024).
> > > >
> > > > [4] Tan, Tao, et al. "PEAR: Position-Embedding-Agnostic Attention Re-weighting Enhances Retrieval-Augmented Generation with Zero Inference Overhead." arXiv preprint arXiv:2409.19745 (2024).

---

> > > > > ### Comment · Reviewer_wMYQ · 2025-08-09
> > > > >
> > > > > Thank you for your response. Given the typos and ambiguous symbols in important formulas and the missing of important baselines (which would require a complete rewrite of the entire experiments and related work sections), I believe this paper requires significant revision for resubmission. Therefore, I will retain my rating.

---

### Official Review · Reviewer_c8Si · 2025-07-01

**Clarity:** 3
**Significance:** 2
**Originality:** 2
**Rating:** 4
**Confidence:** 4

**Summary:**

This paper aims to identify key issues in long-range dependency modeling, particularly in multi-document question answering, and presents some experimental techniques to improve LLM performance targeting these deficiencies. The key problems identified are that language models have difficulty processing information located in the middle of long inputs and struggle to focus on important information.

The authors propose a dual-stage mechanisms to address these issues. First, Contextual Gate Weighting: for each paragraph, the method measures information flow with respect to the question, ranks the paragraphs, and multiplies the attention scores by learned weights that combine contextual relevance and position-aware factors. The second mechanism is Reciprocal Attention Suppression, which aims to mute the attention between key and irrelevant paragraphs by downscaling the cross-attention between them.

The paper runs several experiments on open-source models and multi-document question answering datasets and reports gains in F1 scores in the majority of the settings.

**Questions:**

Could the authors demonstrate that the same process, with the current parameter settings, generalizes to other domains—even on a small toy dataset—to help build confidence in the method’s robustness?

Could the authors provide empirical data on the computational overhead or latency introduced by the proposed method compared to standard attention?

How were the hyperparameters tuned, and what was the data splitting strategy used to avoid potential data leakage or overfitting?

Do the authors have any hypotheses for why the method appears to perform best on medium-sized language models only?

**Ethical Concerns:**

["NO or VERY MINOR ethics concerns only"]

**Final Justification:**

I thank the authors for their detailed rebuttal. The additional results definitely help in clearing most of my apprehensions. I also appreciate the discussion around computational overhead, latency and robustness. Thus, I am happy to raise my score.

**Limitations:**

Please check above for broad feedback, I don’t think this paper has a negative societal impact.

**Quality:**

3

**Strengths And Weaknesses:**

### Strengths

The paper identifies key problems in the question answering field, and I agree that both of these research directions are important for the community.

The proposed method requires minimal engineering and no training runs, so it's quite easy to use.

The language of the paper is clear, and the flow is easy to understand. Attention reweighing has been studied in the community before, but the techniques presented provide an interesting additional reuse case in an experimental setting. The authors have also done a good job citing the relevant previous work in the area.

The F1 improvements, although in some cases quite marginal, are consistent and certainly improve confidence in the method.

### Weaknesses

The method needs deep access to the attention stack and major changes to the inference loop. Although it is training-free, it can only be applied to open-source transformer models with the usual infrastructure. The computational overhead is also not trivial. I am also not sure how the latency will be affected.

Since the paper is completely experimental and the techniques are quite arbitrary without any theoretical justification, the paper needs to clear a very high bar in the evaluation process and present a generalizable method. The current manuscript doesn’t clear that bar satisfactorily. The proposed methods seem too specific to the hyperparameters, which are not practical to tune for every specific task. This prevents the use of this method more generally. The hyperparameter tuning process needs a lot of clarification about the data splitting and tuning process.

As the authors mentioned, for some models like LLaMA (Figure 8), some choices of hyperparameters are worse than the baseline. In other scenarios, a lot of the hyperparameter settings are very close to the baseline. So, it does seem like the precise choice plays a very strong role and is quite model-dependent. “Middle size models” is a bit confusing notion and depends on the model architecture, so it's not clear whether the method will be useful or not on a particular model without having to do some kind of holdout evaluation.

Overall, I agree with the problems identified in the paper, which are both important and well-motivated. However, the proposed solutions come across as somewhat ad hoc and appear to address the symptoms rather than the root causes. While I recognize that tackling the underlying issues is substantially more challenging, the current approach may reflect some degree of research overfit.

---

> ### Author Rebuttal · Authors · 2025-07-31
>
> We sincerely thank the reviewer for the thoughtful comments regarding the computational overhead, theoretical justification, hyperparameter tuning, model generalization, and method design. Based on the weaknesses you identified, we have organized our responses into the following parts:
>
> ***Weaknesses1: Computational Overhead and Latency***
>
> This discussion refers to paragraph 1 in the weaknesses the reviewer proposed.
>
> (i) We acknowledge that DSAS needs deep access to the attention stack, but it is a lightweight plugin and does not require major changes to the inference loop. DSAS intervenes only in specific layers (the final 50 %), where it computes a contextual gate weight for each paragraph based on the raw attention scores and dynamically adjusts those scores. The overall procedure preserves the standard Transformer computations and requires no architectural modifications.
>
> (ii) We acknowledge that DSAS has only been applied to open-source Transformer models. However, this constraint does not imply a technical shortcoming. In fact, existing attention optimization methods are also limited to such architectures.
>
> (iii) Regarding computational overhead, as mentioned in Appendix D, the additional operations of DSAS introduce $\mathcal{O}(LQ)$ time complexity, where $L$ and $Q$ denote the input length and the question length, respectively. Since it scales linearly with $L$, it **constitutes a considerably smaller fraction of the total computation compared to the quadratic $\mathcal{O}(L^2)$ complexity** of standard attention mechanisms.
>
> (iv) To evaluate inference latency, we conducted experiments on Llama-3.1-8B using a single NVIDIA A800 GPU. Table 1 reports the average per-sample inference time for a LongBench subset (L-HotpotQA). Results indicate DSAS introduces a 10% latency increase over the baseline method.  **We contend the modest overhead is worthwhile given DSAS's consistent performance gains across diverse tasks.**
>
> Table 1 Average latency comparisons on L-HotpotQA of Llama-3.1-8B.
>
> |          | Latency (s) |
> | :-: | :-: |
> | Baseline | 2.61        |
> | Ours     | 2.87        |
>
> ***Weaknesses2:  Lack of Theoretical Justification***
>
> This discussion refers to paragraph 2 in the weaknesses the reviewer proposed. We would like to offer the following clarifications on the techniques used in the manuscript:
>
> * Although the validation of DSAS is mainly experimental, the design of DSAS is firmly **grounded in an information-theoretic perspective on attention** rather than ad hoc heuristics. In Section 2, we analyze information flows at both paragraph disparity and answer quality levels, and these insights ​directly provide the basis​ for our CGW and RAS modules.
> * The used techniques align with our research goal. e.g., we adopt **Gaussian CDF to compute positional values** for enhancing the model's focus on centrally located paragraphs, due to the ''U‑shaped attention bias'' [1] in LLMs.
> * We designed DSAS as a truly plug-and-play mechanism whose generalizability is empirically validated by numerous experiments. **DSAS derives consistent improvements across diverse LLMs and tasks**. In addition, in response to ***Q1***, we have extended DSAS to additional tasks, including summarization and code completion, further demonstrating its generalizability.
>
> [1] Hsieh, Cheng-Yu, et al. "Found in the middle: Calibrating positional attention bias improves long context utilization." arXiv preprint arXiv:2406.16008 (2024).
>
> ***Weaknesses3:  Details of Hyperparameter Tuning***
>
> This discussion refers to paragraph 2 in the weaknesses the reviewer proposed.
>
> (i) The hyperparameter configuration "$K$=10, $n$=50%, $\alpha$=1.0, $\beta$=0.7" is effective across diverse LLMs on various tasks (4.2% average gains on Llama-3.1-8B and Qwen2.5-14B). Our work includes the design of DSAS as well as the finding of the robust hyperparameter configuration. We believe the robustness benefits from the well-designed formulations and there is no need for hyperparameter tuning process.
>
> (ii) In response to ***Q1***, we have extended DSAS to additional tasks, including summarization and code completion. Table 1 confirms that the hyperparameter configuration "$K$=10, $n$=50%, $\alpha$=1.0, $\beta$=0.7" maintains **consistent efficacy across these unseen domains.**
>
> ***Weaknesses4:  Generalization across Models***
>
> This discussion refers to paragraph 3 in the weaknesses the reviewer proposed. We have discussed some points of the weakness. Please refer to the response to ***Weaknesses3*** above.
>
> (i) We acknowledge that LLM+DSAS underperforms the baseline under some hyperparameter settings on Llama-3.2-3B. However, this does not indicate instability of the method itself. Rather, it reflects the inherent limitations of small‑scale models, which we attribute to their relatively weaker semantic comprehension capacity and greater sensitivity to input noise, which requires more precise hyperparameter tuning. **In contrast, mainstream 7B–14B models exhibit significantly lower sensitivity to these hyperparameters (Figures 5, 9, 10).**
>
> (ii) Medium-sized LLMs refer to LLMs with parameter sizes ranging from 7B to 14B, regardless of architectural family (e.g., LLaMA-3.1-8B, Qwen2.5-7B, Mistral-7B, Qwen2.5-14B).
>
> ***Weaknesses5:  Method Design***
>
> This discussion refers to paragraph 4 in the weaknesses the reviewer proposed.
>
> (i) As noted in Introduction, existing approaches either sacrifice global token interaction or require additional training, which limits their adaptability. In contrast, DSAS does not reflect research overfit and is founded on a systematic analysis of information flows to guide attention optimization.
>
> (ii) In response to ***Weaknesses2***, we have provided a detailed explanation of the theoretical basis of DSAS.
>
> (iii) In response to ***Weaknesses1,3,4***, quantitative results confirm performance gains across diverse tasks and models while maintaining inference efficiency. The strong capability enhancement indicates DSAS addresses root causes rather than symptoms.
>
> ***Q1: Robustness of DSAS***
>
> We appreciate the reviewer's suggestion to evaluate DSAS's scalability beyond Multi‑doc QA. Accordingly, we applied DSAS to the summarization (GovReport, QMSum, MultiNews) and code completion (LCC, RepoBench-P) tasks in the LongBench benchmark. After examining the prompt templates, we selected the final instruction sentence as the anchor (e.g., "Now, write a one‑page summary of the report." for GovReport; "Next line of code:" for LCC).
>
> Because DSAS requires paragraph‑level segmentation and Multi‑doc QA naturally provides this, we instead segment these new tasks by fixed token counts. Specifically, we use 500‑token segments for GovReport, QMSum, and RepoBench‑P, and 200‑token segments for MultiNews and LCC, ensuring a medium number (5-40) of paragraphs per example. The comparison results are shown in Table 2.
>
> We observe: (a) DSAS achieves consistent performance gains **across ​​all five tasks across three medium-size model architectures​​.** (b) The positive results **extend to diverse tasks (summarization and code completion)**, confirming that DSAS's benefits are ​​not confined to Multi-doc QA​​.
>
> Table 2 Comparison results on summarization and code completion tasks with "$K$=10, $n$=50%, $\alpha$=1.0, $\beta$=0.7". The evaluation metrics for L-GovReport, L-QMSum, L-MultiNews and L-LCC, L-RBP are Rouge-L score and Edit Sim, respectively. RBP denotes RepoBench-P.
>
> | Models       | Method   | L-GovReport | L-QMSum  | L-MultiNews | L-LCC    | L-RBP    | Average  |
> | :-: | :-: | :-: | :-: | :-: | :-: | :-: | :-: |
> | Qwen-2.5-7B  | Baseline | 33.6        | 22.4     | 23.7        | 53.7     | 48.2     | 36.3     |
> |              | Ours     | **37.0**    | **25.1** | **26.2**    | **56.3** | **51.5** | **39.2** |
> | Llama-3.1-8B | Baseline | 34.9        | 24.8     | 27.1        | 58.1     | 50.8     | 39.1     |
> |              | Ours     | **37.4**    | **26.6** | **29.3**    | **60.0** | **52.5** | **41.2** |
> | Qwen-2.5-14B | Baseline | 38.4        | 28.2     | 31.8        | 63.9     | 55.6     | 43.6     |
> |              | Ours     | **41.0**    | **30.4** | **34.6**    | **66.7** | **58.9** | **46.3** |
>
> ***Q2: Computational Overhead and Latency***
>
> We have discussed this question. Please refer to the response to ***Weaknesses1*** above.
>
> ***Q3: Hyperparameter Tuning and Data Splitting Strategy***
>
> We have discussed this question. Please refer to the response to ***Weaknesses4*** above.
>
> Furthermore, to prevent data leakage, all experiments are conducted on the validation sets of HotpotQA, 2WikiMultiHopQA, MuSiQue datasets. For LongBench datasets, we tested all samples following the common practice.
>
> ***Q4: Hypotheses for the Best Performance on Medium-sized Language Models***
>
> (i) Hypothesis 1: Balance between comprehension capacity and optimization potential.
>
> Medium-sized LLMs (e.g., Llama-3.1-8B, Qwen2.5-14B) demonstrate adequate semantic understanding yet remain vulnerable to input noise, especially in long-context scenarios. DSAS overcomes this by analyzing information flows to precisely identify critical paragraphs and amplify model focus. In contrast, smaller models (e.g., Llama-3.2-3B) lack sufficient comprehension capacity, while larger models (e.g., Qwen2.5-32B) approach the task performance ceiling, leaving little room for further gains.
>
> (ii) Hypothesis 2: Dependence on positional awareness.
>
> Medium-sized LLMs exhibit greater vulnerability to the "lost-in-the-middle" phenomenon. DSAS mitigates this through its position-aware weighting $g\_m$. The experiments of ablation study reveal that removing $g\_m$ causes a marked performance drop for Qwen2.5-7B and Llama-3.1-8B, whereas Qwen2.5-14B's exhibits a smaller decline.

---

> > ### Comment · Reviewer_c8Si · 2025-08-06
> >
> > I thank the authors for their detailed rebuttal. The additional results definitely help in clearing some of my apprehensions. I also appreciate the discussion around computational overhead, latency and robustness. Thus, I am happy to raise my score.

---

> > > ### Author Response · Authors · 2025-08-06
> > >
> > > We thank the reviewer again, for the discussion and for raising the score. We sincerely appreciate the reviewer's valuable time and suggestions, which helped us to improve the quality of this work.

---

> > ### Public Comment · ~Yujie_Jin1 · 2025-11-18
> > **Can the method adapt to vLLM inference framework？**
> >
> > Hi, I have been impressed by the innovative method proposed in your paper. Recently, we have been attempting to extract attention scores during the vLLM - based inference phase for attribution and interpretability research, but we encountered difficulties—vLLM does not seem to support providing attention scores (though I might have missed something).  Since your method also needs deep access to attention scores, I am curious whether this method can be adapted to the vLLM inference framework? It would be of great help to my understanding if you could give a brief explanation.

---

### Official Review · Reviewer_Gbi5 · 2025-07-01

**Clarity:** 3
**Significance:** 2
**Originality:** 2
**Rating:** 3
**Confidence:** 4

**Summary:**

This paper introduces Dual-Stage Adaptive Sharpening (DSAS), a training-free framework designed to optimize attention in large language models (LLMs) for multi-document question answering. The DSAS framework consists of two main modules: Contextual Gate Weighting (CGW), which weighs paragraph relevance through attention tracking and positional weighting, and Reciprocal Attention Suppression (RAS), which suppresses unnecessary attention between key and irrelevant paragraphs. The method is applicable as a plug-and-play module requiring no architectural modification or extra parameters. The paper presents systematic analysis of information flow in LLMs and validates DSAS across four public benchmarks and several mainstream LLM architectures.

**Questions:**

1. The position-aware weighting, aimed at the "lost-in-the-middle" problem, may excessively prioritize central paragraphs even when relevant evidence occurs at the edges. While the authors mention mitigation via ranking and capping, additional discussion on failure cases of this bias (especially in non-standard input orderings) is lacking.
2. DSAS is only evaluated on Multi-doc QA tasks, whereas LLMs are also deployed for other long-context applications like summarization. It is suggested to conduct experiments on other tasks. Moreover, the proposed method is not evaluated over general performance benchmarks, such as how DSAS affects math/code or other reasoning tasks.
3. Figure 5 reveals non-negligible variance in gains depending on choices of K, n, α, β. These variances indicate the proposed method is unstable and could be hard to generalize to unseen domains.

**Ethical Concerns:**

["NO or VERY MINOR ethics concerns only"]

**Limitations:**

see above.

**Quality:**

3

**Strengths And Weaknesses:**

1. This paper is well-written and well-organized. The idea is simple and easy to follow, and is built upon in-depth analysis of layer-wise information flow.
2. The ablations (Table 3) and hyperparameter studies (Figure 5) dissect the individual contributions of CGW, RAS, and positional weighting.

---

> ### Author Rebuttal · Authors · 2025-07-31
>
> We acknowledge the reviewer's valuable insights regarding potential paragraph ordering sensitivity, DSAS's scalability, and hyperparameter configurations.
>
> ***Q1: Experiments of Non-standard Orderings***
>
> As DSAS is designed to improve attention toward centrally located paragraphs, thereby mitigating the LLMs' inherent ''lost‑in‑the‑middle'' issue. It is important to evaluate DSAS's robustness to input order. Accordingly, we conducted an experiment where we randomized the ordering of paragraphs in all test samples while deliberately increasing the probability of supporting paragraphs occurring at the edges. Table 1 presents results for both original and shuffled inputs, using the consistent configuration "$K$=10, $n$=50%, $\alpha$=1.0, $\beta$=0.7". **Results indicate minor performance fluctuations across ordering variants.** This stability demonstrates DSAS's robustness to varying input orders, primarily benefits from our weighting strategy. We think that when evidence appears at the edges​ (where $g\_m$ in Equation (9) is reduced)​, DSAS attenuates the adjustments to the model's native attention distribution. This design intentionally leverages the model's inherent capacity​​ to identify and amplify information flows from the evidence at the edges, which aligns with the ''U-shaped attention bias'' phenomenon in [1].
>
> We observed one case in MuSiQue with three supporting paragraphs (evaluated using Llama-3.1-8B) that succeeded with the original ordering but failed after shuffling. Analysis revealed that shuffling positioned the supporting paragraphs more centrally. **The observed failure in this instance may stem from insufficient focus amplification on central information despite applying DSAS.** This is supported by our information flow metric $\frac{1}{Q}\mathcal{I}\_{p^s,q}+\mathcal{I}\_{p^s,t}$, where the shuffled sample is 27.75 compared to 29.87 for the original. Nevertheless, DSAS maintains robust performance across the vast majority of samples.
>
> Table 1 Results of original and shuffled input orderings.
>
> | Models       | Method   | HotpotQA | 2WikiMultiHopQA | MuSiQue  | L-HotpotQA | L-2WikiMultiHopQA | L-MuSiQue |
> | :-: | :-: | :-: | :-: | :-: | :-: | :-: | :-: |
> | Qwen-2.5-7B  | Original | **46.1** | **49.9**        | 35.0     | **57.7**   | 52.7              | **33.4**  |
> |              | Shuffled | 46.1     | 49.7            | **35.4** | 57.6       | **53.0**          | 33.3      |
> | Llama-3.1-8B | Original | 47.1     | 50.8            | **39.2** | **56.5**   | 47.3              | **32.0**  |
> |              | Shuffled | **47.2** | **51.1**        | 39.0     | 56.2       | **47.5**          | 31.8      |
> | Qwen-2.5-14B | Original | **51.8** | 58.2            | **43.8** | **60.9**   | **56.1**          | **39.3**  |
> |              | Shuffled | 51.7     | **58.3**        | 43.8     | 60.9       | 56.0              | 39.3      |
>
> [1] Hsieh, Cheng-Yu, et al. "Found in the middle: Calibrating positional attention bias improves long context utilization." arXiv preprint arXiv:2406.16008 (2024).
>
> ***Q2: Performance of Other Tasks***
>
> We appreciate the reviewer's suggestion to evaluate DSAS's scalability beyond Multi‑doc QA. Accordingly, we applied DSAS to the summarization (GovReport, QMSum, MultiNews) and code completion (LCC, RepoBench-P) tasks in the LongBench benchmark. After examining the prompt templates, we selected the final instruction sentence as the anchor (e.g., "Now, write a one‑page summary of the report." for GovReport; "Next line of code:" for LCC).
>
> Because DSAS requires paragraph‑level segmentation and Multi‑doc QA naturally provides this, we instead segment these new tasks by fixed token counts. Specifically, we use 500‑token segments for GovReport, QMSum, and RepoBench‑P, and 200‑token segments for MultiNews and LCC, ensuring a medium number (5-40) of paragraphs per example. The comparison results are shown in Table 2.
>
> We observe: (a) DSAS achieves consistent performance gains **across ​​all five tasks across three medium-size model architectures​​.** (b) The positive results **extend to diverse tasks (summarization and code completion)**, confirming that DSAS's benefits are ​​not confined to Multi-doc QA​​.
>
> Table 2 Comparison results on summarization and code completion tasks with "$K$=10, $n$=50%, $\alpha$=1.0, $\beta$=0.7". The evaluation metrics for L-GovReport, L-QMSum, L-MultiNews and L-LCC, L-RBP are Rouge-L score and Edit Sim, respectively. RBP denotes RepoBench-P.
>
> | Models       | Method   | L-GovReport | L-QMSum  | L-MultiNews | L-LCC    | L-RBP    | Average  |
> | :-: | :-: | :-: | :-: | :-: | :-: | :-: | :-: |
> | Qwen-2.5-7B  | Baseline | 33.6        | 22.4     | 23.7        | 53.7     | 48.2     | 36.3     |
> |              | Ours     | **37.0**    | **25.1** | **26.2**    | **56.3** | **51.5** | **39.2** |
> | Llama-3.1-8B | Baseline | 34.9        | 24.8     | 27.1        | 58.1     | 50.8     | 39.1     |
> |              | Ours     | **37.4**    | **26.6** | **29.3**    | **60.0** | **52.5** | **41.2** |
> | Qwen-2.5-14B | Baseline | 38.4        | 28.2     | 31.8        | 63.9     | 55.6     | 43.6     |
> |              | Ours     | **41.0**    | **30.4** | **34.6**    | **66.7** | **58.9** | **46.3** |
>
> ***Q3: Hyperparameter Settings***
>
> (i) As demonstrated in Figure 5 and Appendix B, we evaluated the impact of hyperparameters $K, n, \alpha, \beta$ across multiple models. While performance exhibits fluctuations with different hyperparameter configurations, the vast majority of settings still outperform the baseline. Notably, substantial variability occurs only with the smaller Llama‑3.2‑3B model (Figure 8), which we attribute to its relatively weaker semantic comprehension capacity and greater sensitivity to input noise, which requires more precise hyperparameter tuning. In contrast, **mainstream 7B–14B models exhibit significantly lower sensitivity to these hyperparameters.**
>
> (ii) Section 4.2 confirms that the fixed hyperparameter combination "$K$=10, $n$=50%, $\alpha$=1.0, $\beta$=0.7" demonstrates stable and effective performance. The results highlight two properties of DSAS: (a) **Model-agnostic**: It consistently improves performance across diverse architectures (Llama, Qwen, and Mistral) ranging from 3 B to 32 B parameters. (b) **Task-agnostic**: It yields uniform improvements across multiple tasks. Furthermore, Figures 2 and 6 illustrate that information flows from supporting and negative paragraphs diverge substantially in deeper layers (last 50 %), whereas shallow layers exhibit limited divergence. Figure 5 and Appendix B further confirm that $n$=50% is the optimal choice, reinforcing the robustness of the hyperparameters.
>
> (iii) In response to **Q2**, we have extended DSAS to additional tasks, including summarization and code completion. Supplementary results in Table 2 confirm that the hyperparameter configuration "$K$=10, $n$=50%, $\alpha$=1.0, $\beta$=0.7" maintains consistent efficacy **across these unseen domains.**

---

### Official Review · Reviewer_KV6y · 2025-07-05

**Clarity:** 3
**Significance:** 3
**Originality:** 3
**Rating:** 5
**Confidence:** 4

**Summary:**

This paper presents DSAS, a universal plug-and-play framework for attention optimization in multi-document question answering. The goal is to improve the long-context understanding ability of large language models in multi-document question answering tasks by introducing some inductive biases for such tasks. Concretely, Contextual Gate Weighting (CGW) and Reciprocal Attention Suppression (RAS) are proposed and combined to address this problem. Extensive experiments on four benchmarks demonstrate DSAS's universality across mainstream LLMs.

**Questions:**

- How are the performance of other tasks if DSAS is applied to the LLMs? Will the performance significantly decrease, or comparable to the baseline without DSAS?
- Can DSAS be used for context length extension? For example, suppose that a model is trained on 128K context-length, what is the performance if we apply DSAS on it to process 256K-tokens multi-document question answering tasks?

**Ethical Concerns:**

["NO or VERY MINOR ethics concerns only"]

**Final Justification:**

The authors have well addressed my concerns. I would like to keep my positive rating.

**Limitations:**

YES

**Quality:**

3

**Strengths And Weaknesses:**

Pros:
- This paper is well-written and easy to understand. The problem is clearly presented, and the motivation is well discussed. The long-context understanding ability of LLM is a very important research topic in the community, and this paper focuses on a big and typical series of subtasks (multi-document question answering tasks) in this topic.
- The paper is well organized, especially the logic of motivation->method->evaluation and analysis. Section 2, information flow analysis on multi-doc QA, identifies key factors for effective multi-doc QA reasoning in a systematic and quantitative manner, making the proposed methods promising and intuitive.
- The method is lightweight: it doesn't require further training on the LLMs, but just changes how the attention is calculated, based on the properties of multi-document question answering tasks.

Cons:
- It would be better if the experiment and analysis section can include some case study or quantitative analysis to prove that the inherent discrimination capability is really enhanced.
- I'd suggest adding some experiments to further analysis DSAS on different subsets. For example, how will the number of supporting/negative/total paragraphs affect the effectiveness of DSAS?

---

> ### Author Rebuttal · Authors · 2025-07-31
>
> We thank the reviewers for their insightful questions regarding the proofs for the discrimination capability, the need for further analysis on paragraph number, the scalability of DSAS on other tasks, and our applicability for context length extension.
>
> ***Cons1: Proofs for the Enhanced Discrimination Capability***
>
> (i) **We conducted an error analysis of DSAS by examining the confusion matrix visualization in Appendix C.** Figure 11 illustrates the global attention matrix among supporting/negative paragraphs, questions, and target. DSAS substantially enhances the LLM's focus on supporting paragraphs, demonstrating improved discrimination capability.
>
> (ii) In addition, to complement our findings, we have added a quantitative analysis. Specifically, Equations (1) and (2) define the information flows from the m‑th paragraph to the question and to the target, respectively. Building on these definitions, we introduce two metrics: the supporting paragraph gain ratio $\Delta \mathcal{I}\_{p^s}$ and the negative paragraph suppression ratio $\Delta \mathcal{I}\_{p^n}$, which are formulated as:
> $$
> \begin{align}
> \Delta \mathcal{I}\_{p^s}&=\frac{\left(\frac{1}{Q}\mathcal{I}\_{p^s,q}^{'}+\mathcal{I}\_{p^s,t}^{'}\right)-\left(\frac{1}{Q}\mathcal{I}\_{p^s,q}+\mathcal{I}\_{p^s,t}\right)}{\frac{1}{Q}\mathcal{I}\_{p^s,q}+\mathcal{I}\_{p^s,t}}, \\\\
> \Delta \mathcal{I}\_{p^n}&=1-\frac{\frac{1}{Q}\mathcal{I}\_{p^n,q}^{'}+\mathcal{I}\_{p^n,t}^{'}}{\frac{1}{Q}\mathcal{I}\_{p^n,q}+\mathcal{I}\_{p^n,t}},
> \end{align}
> $$
> where $Q$ denotes the question token length; $\mathcal{I}\_{\cdot,q}^{'}$ and $\mathcal{I}\_{\cdot,t}^{'}$ denote the information flows from supporting/negative paragraphs to the question and target after using DSAS, respectively. We calculate the information flows and metrics, and observe that **LLM+DSAS evidently increases the information flows from supporting paragraphs while suppressing information flows from negative ones compared to the baseline LLM.** The results are shown in Table 1.
>
> Table 1 Results of information flows and metrics on HotpotQA, 2WikiMultiHopQA and MuSiQue of Llama-3.1-8B.
>
> |                 | $\frac{1}{Q}\mathcal{I}\_{p^s,q}+ \mathcal{I}\_{p^s,t}$ | $\frac{1}{Q}\mathcal{I}\_{p^n,q}+ \mathcal{I}\_{p^n,t}$ | $\frac{1}{Q}\mathcal{I}\_{p^s,q}^{'}+\mathcal{I}\_{p^s,t}^{'}$ | $\frac{1}{Q}\mathcal{I}\_{p^n,q}^{'}+ \mathcal{I}\_{p^n,t}^{'}$ | $\Delta \mathcal{I}\_{p^s}$ | $\Delta \mathcal{I}\_{p^n}$ |
> | :-: | :-: | :-: | :-: | :-: | :-: | :-: |
> | HotpotQA        | 30.36 | 11.94| 36.21| 8.47| 19.3% | 29.1% |
> | 2WikiMultiHopQA | 30.85 | 12.30| 38.48| 7.94| 24.7% | 35.4% |
> | MuSiQue         | 25.07 | 8.65 | 30.11| 6.52| 20.1% | 24.6% |
>
> (iii) Finally, we selected one example from MuSiQue and **visualized its layer-wise attention matrices** for both the baseline LLM and LLM+DSAS. The matrices for LLM+DSAS exhibit lower attention dispersion [1], indicating its ability to concentrate on consistent token positions within supporting paragraphs. In contrast, the baseline LLM fails to filter information flows from negative paragraphs.
>
> [1] Qin, Ziran, et al. "Cake: Cascading and adaptive kv cache eviction with layer preferences." arXiv preprint arXiv:2503.12491 (2025).
>
> ***Cons2: Further Analysis on Different Subsets***
>
> We have conducted a further analysis on different subsets by changing the number of supporting/negative/total paragraphs.
>
> (i) Since the MuSiQue dataset contains 2-4 supporting paragraphs per sample, we have conducted experiments on subsets grouped by supporting paragraph count. Table 2 presents the comparative performance. **We observe minimal performance gaps with varying paragraph counts**, demonstrating that the baseline LLM is able to answer the questions and DSAS better capture relevant information flows to generate more accurate answers.
>
> Table 2 Results of different supporting paragraph count setting on MuSiQue through the F1-score (%).
>
> | Models       | Method   | #Supporting=2 | #Supporting=3 | #Supporting=4 |
> | :-: | :-: | :-: | :-: | :-: |
> | Qwen-2.5-7B  | Baseline | 30.4          | 30.9          | 30.3          |
> |              | Ours     | **35.0**      | **35.2**      | **34.7**      |
> | Llama-3.1-8B | Baseline | 35.8          | 32.9          | 33.6          |
> |              | Ours     | **40.2**      | **35.7**      | **36.5**      |
> | Qwen-2.5-14B | Baseline | 38.1          | 37.3          | 38.7          |
> |              | Ours     | **44.0**      | **43.3**      | **44.3**      |
>
> (ii) Next, to examine the effect of total paragraph count, we randomly sampled 200 examples from the MuSiQue dataset and constructed inputs with 10, 20, 30, and 40 paragraphs. Since each example originally contains 20 paragraphs, the 20‑paragraph setting uses the original sample inputs. For the 10‑paragraph setting, we randomly removed ten of the negative paragraphs. For the 30‑paragraph and 40‑paragraph settings, we randomly adding 10/20 paragraphs to construct inputs, respectively, drawn from other MuSiQue samples. The results of varying total paragraph count are reported in Table 3.
>
> The results indicate that: (a) As the total paragraph count increases (longer input sequences), performance generally declines. However, **LLM + DSAS exhibits a slower degradation trend**, as it effectively suppresses information flows from negative paragraphs. (b) **Llama‑3.1‑8B + DSAS even outperforms Qwen‑2.5‑14B under all settings**, demonstrating the effectiveness of DSAS as a plug-and-play strategy.
>
> Table 3 Results of different total paragraph count setting on MuSiQue through the F1-score (%).
>
> | Models       | Method   | #Total=10 | #Total=20 | #Total=30 | #Total=40 |
> | :-: | :-: | :-: | :-: | :-: | :-: |
> | Qwen-2.5-7B  | Baseline | 32.6      | 30.7      | 30.2      | 29.5      |
> |              | Ours     | **36.8**  | **35.1**  | **34.8**  | **34.4**  |
> | Llama-3.1-8B | Baseline | 35.4      | 34.8      | 33.9      | 32.3      |
> |              | Ours     | **39.7**  | **39.3**  | **39.0**  | **38.1**  |
> | Qwen-2.5-14B | Baseline | 39.8      | 38.4      | 37.1      | 36.6      |
> |              | Ours     | **44.9**  | **44.0**  | **43.4**  | **42.7**  |
>
> ***Q1: Performance of Other Tasks***
>
> We appreciate the reviewer's suggestion to evaluate DSAS's scalability beyond Multi‑doc QA. Accordingly, we applied DSAS to the summarization (GovReport, QMSum, MultiNews) and code completion (LCC, RepoBench-P) tasks in the LongBench benchmark. After examining the prompt templates, we selected the final instruction sentence as the anchor (e.g., "Now, write a one‑page summary of the report." for GovReport; "Next line of code:" for LCC).
>
> Because DSAS requires paragraph‑level segmentation and Multi‑doc QA naturally provides this, we instead segment these new tasks by fixed token counts. Specifically, we use 500‑token segments for GovReport, QMSum, and RepoBench‑P, and 200‑token segments for MultiNews and LCC, ensuring a medium number (5-40) of paragraphs per example. The comparison results are shown in Table 4.
>
> We observe: (a) DSAS achieves consistent performance gains **across ​​all five tasks across three medium-size model architectures.** (b) The positive results **extend to diverse tasks (summarization and code completion)**, confirming that DSAS's benefits are ​​not confined to Multi-doc QA​​.
>
> Table 4 Comparison results on summarization and code completion tasks with "$K$=10, $n$=50%, $\alpha$=1.0, $\beta$=0.7". The evaluation metrics for L-GovReport, L-QMSum, L-MultiNews and L-LCC, L-RBP are Rouge-L score and Edit Sim, respectively. RBP denotes RepoBench-P.
>
> | Models       | Method   | L-GovReport | L-QMSum  | L-MultiNews | L-LCC    | L-RBP    | Average  |
> | :-: | :-: | :-: | :-: | :-: | :-: | :-: | :-: |
> | Qwen-2.5-7B  | Baseline | 33.6        | 22.4     | 23.7        | 53.7     | 48.2     | 36.3     |
> |              | Ours     | **37.0**    | **25.1** | **26.2**    | **56.3** | **51.5** | **39.2** |
> | Llama-3.1-8B | Baseline | 34.9        | 24.8     | 27.1        | 58.1     | 50.8     | 39.1     |
> |              | Ours     | **37.4**    | **26.6** | **29.3**    | **60.0** | **52.5** | **41.2** |
> | Qwen-2.5-14B | Baseline | 38.4        | 28.2     | 31.8        | 63.9     | 55.6     | 43.6     |
> |              | Ours     | **41.0**    | **30.4** | **34.6**    | **66.7** | **58.9** | **46.3** |
>
> ***Q2: DSAS for Context Length Extension***
>
> (i) Our proposed DSAS aims to address two challenges in Multi‑doc QA: limited long-range dependency and persistent ''lost-in-the-middle'' issue. Although DSAS has not been extended in scenarios exceeding the fixed window size, it offers a plug‑and‑play solution that enhances discrimination without architecture modifications.
>
> (ii) In future work, we plan to extend DSAS beyond the fixed context window by integrating it with techniques such as StreamingLLM [2], which enlarges effective context length by retaining the KV cache of initial tokens and the most recent tokens. Specifically, we will investigate methods for dynamically preserving information from key paragraphs while discarding irrelevant noise during generation, thereby supporting substantially longer contexts.
>
> [2] Xiao, Guangxuan, et al. "Efficient streaming language models with attention sinks." arXiv preprint arXiv:2309.17453 (2023).
>
> We will incorporate all the analysis and supplementary experiments into the revised manuscript.

---

### Comment · Area_Chair_obrS · 2025-08-07
**Please participate in discussions with authors before submitting “Mandatory Acknowledgement”**

Dear Reviewers,

I see that some of you have not yet responded to the author's rebuttal with further feedback. Please make sure to do so, which is important to reach a well thought out  consensus for the quality of this paper.

Best regards,
Submission20041 AC

---

### Note · Authors · 2025-08-13

Dear reviewers and chairs,

We thank all reviewers and AC for their time and constructive feedback. Below, we summarize the consensus on our work and our clarifications regarding the key concerns.

**Recognized Strengths**

- **Clear method novelty**: We introduce DSAS, a training-free, plug-and-play attention optimization method to address two critical issues in Multi-doc QA.
- **Consistent performance gains**: DSAS achieves stable improvements across all Multi-doc QA tasks and diverse LLMs, advancing research in LLM generation.
- **Well-organized paper structures**: The work is organized with clear motivation, systematic information flow analysis, and sufficient experimental validation.

**Clarifications on Concerns**

- **Generalization and robustness**: To KV6y, Gbi5 and c8Si, added results confirm that DSAS can be applied to other tasks. To Gbi5 and c8Si, we clarify the robustness of our method about hyper-parameter tuning.
- **Analysis of different inputs**: To KV6y, we perform analysis with varying paragraph counts. To Gbi5, we conduct experiments with non-standard input orderings. These results further demonstrate that our method is well-designed and robust.
- **Computational overhead**: To c8Si, we prove DSAS only introduces an acceptable 10% latency increase.
- **Performance comparisons**: To wMYQ, we compare DSAS with PINE, AttentionRag, SPHS, and PEAR, and the results show that DSAS outperforms them.

**Remaining Discussion**

Lastly, we'd like to emphasize some remaining problems in the final response by wMYQ.

(1) Experiments need rewriting: Gbi5 and c8Si acknowledge the ablation study and the consistent performance gains, respectively, demonstrating the completeness of our evaluation and the effectiveness of DSAS. However, the experiments suggested by wMYQ are valuable in providing further supporting evidence for DSAS. We conduct the proposed experiments, which further corroborate our results, and some of the findings will be included in the appendix of a later version.

(2) Related works need rewriting: Some attention manipulation-based works focus on mitigating ''lost-in-the-middle'' and they do improve generation abilities. We concede the overlook of such works since we focus on improving the performance specifically on Multi-doc QA. Other relevant methods which truncate long contexts are inspired by RAG, and we have conducted a good review of them, which has also been acknowledged by c8Si. Therefore, only minor revisions are required.

---

### Decision · Program_Chairs · 2025-09-17

**Decision:**

Accept (poster)

**Comment:**

This paper studies the long-range dependency modeling problem, particularly in multi-document question answering. To mitigate the  long texts issue and ''lost-in-the-middle'' issue the authors propose a dual-stage mechanism which first, ranks the paragraphs, and multiplies the attention scores by learned weights and then mute the attention between key and irrelevant paragraphs by downscaling the cross-attention between them. Several experiments on open-source models and multi-document question answering datasets and reports gains in F1 scores in the majority of the settings. The experiment was later expanded to 2 other tasks in the LongBench benchmark (coding, summarizations).

Reviewers recognise that this is a key challenge in  the field of LLMs and question answering. They also recognize the simplicity of the approach (no post-training) and consistent improvements in the quality. Reviewers also suggested areas to improve such as
1) theoretical justification of the method
2) discussions of overhead of the method and failure modes.